# Investigation Into Boundary Layer Transition Using Wall-Resolved LES and Modeled Inflow Turbulence

Brandon Arthur Lobo[1], Alois Peter Schaffarczyk[1], and Michael Breuer[2]

[1]Mechanical Engineering Department, Kiel University of Applied Sciences, D-24149 Kiel, Germany
[2]Professur für Strömungsmechanik, Helmut-Schmidt-Universität Hamburg, D-22043 Hamburg, Germany

**Correspondence:** B. A. Lobo: brandon.a.lobo@fh-kiel.de

**Abstract.** The objective of the present paper is to investigate the transition scenario of the flow around a typical section of a wind turbine blade exposed to different levels of inflow turbulence. A rather low Reynolds number of $\mathrm{Re}_c = 10^5$ is studied at a fixed angle of attack but under five different turbulence intensities ($TI$) up to $TI = 11.2$ %. Using wall-resolved large-eddy simulations combined with an inflow procedure relying on synthetically generated turbulence and a source-term formulation for its injection within the computational domain, relevant flow features such as the separation bubble, inflectional instabilities and streaks can be investigated. The study shows that the transition scenario significantly changes with rising $TI$, where the influence of inflectional instabilities due to an adverse pressure gradient decreases, while the influence of streaks increases resulting in a shift from the classical scenario of natural transition to bypass transition. The primary instability mechanism in the separation bubble is found to be inflectional and its origin is traced back to the region upstream of the separation. Thus, the inviscid inflectional instability of the separated shear layer is an extension of the instability of the attached adverse pressure gradient boundary layer observed upstream. The boundary layer is evaluated to be receptive to external disturbances such that the initial energy within the boundary layer is proportional to the square of the turbulence intensity. Boundary layer streaks were found to influence the instantaneous separation location depending on their orientation. A varicose mode of instability is observed on the overlap of the leading edge of a high-speed streak with the trailing edge of a low-speed streak. The critical amplitude of this instability was analyzed to be about 32 % of the free-stream velocity.

## 1 Introduction

Rotor blades are the determining component for both performance and loads of wind turbines and are therefore key components of further optimizations. To obtain high efficiencies (Schaffarczyk, 2014) an increased use of special aerodynamic profiles (Jones, 1990) is observed possessing large areas of low-resistance, which means laminar flow is maintained. In order to design such profiles, it is necessary to include the laminar-turbulent transition in CFD simulations of wind turbine blades.

To date, there is no CFD model that accurately predicts the location of laminar-turbulent transition under a wide range of operating conditions. A decisive ingredient of these operating conditions is atmospheric inflow turbulence with varying parameters such as the turbulence intensity ($TI$), the length and time scales and the anisotropy. Transition to turbulence at low $TI$ levels under 0.5 % typically takes place through the growth of two-dimensional Tollmien-Schlichting (T-S) distur-

bances that develop three-dimensional, non-linear secondary instabilities which eventually break down to fully developed turbulence (Reshotko, 1976, 2001). At higher $TI$ levels of about 1 % or more, it has been observed experimentally that transition to turbulence bypasses one or more of the typical pre-transitional events occurring through the natural transition route. Bypass transition, a term coined by Morkovin (1969), is then said to have taken place.

Jacobs and Durbin (2001) conducted a direct numerical simulation (DNS) to study bypass transition in an initially laminar
boundary layer with zero mean pressure gradient exposed to numerically simulated free-stream turbulence (FST). At this point it was already well established that in response to forcing by FST a laminar boundary layer develops high-amplitude, low-frequency perturbations referred to as Klebanoff modes, streamwise jets or streaks (Klebanoff et al., 1962). Jacobs and Durbin (2001) found that transition precursors consist of long backward jets contained in the fluctuating streamwise velocity field, i.e., they are directed backwards relative to the local mean streamwise velocity. It must be noted that the total velocity is
not reversed in these structures. Some of these jets extend into the upper region (lift-up mechanism) of the boundary layer and interact with the free-stream perturbations to develop turbulent spots which spread longitudinally and laterally before ultimately merging with the downstream turbulent boundary layer. This lift-up mechanism was first observed as turbulent bursts by Kline et al. (1967) and early studies were conducted, for example, by Landahl (1975, 1980), where the physical mechanism for the amplification of streaks was attributed to the displacement of mean momentum or lift-up. Shortly before the study by Jacobs
and Durbin, a similar phenomenon was observed by Wu et al. (1999) in the transition beneath periodically passing wakes where elongated streaks were precluded due to the finite wake width, but shorter puffs were seen. Jacobs and Durbin (2001) proposed that backward jets seem to be a link between free-stream eddies and the boundary layer, at least in cases with zero mean pressure gradient.

The lift-up mechanism described above is one of the many mechanisms that contribute to boundary layer transition. Plenty
of research has been conducted on bypass transition within boundary layers and beneath vortical disturbances. Various factors have been found to govern the dominant mechanism, for instance: In the absence of a leading-edge and notable pressure gradient effects as in the study by Jacobs and Durbin (2001), bypass transition proceeds mainly through the amplification of Klebanoff distortions. Kendall (1991) observed that the growth of T-S waves was influenced by the leading-edge geometry. While low $TI$ levels influenced the location of transition, no noticeable influence on the amplitude of the Klebanoff distortions
was reported.

Studies by Abu-Ghannam and Shaw (1980) and Gostelow et al. (1994) have found that the past history of the pressure gradient or the mean pressure gradient affects transition rather than the local value at transition. Nagarajan et al. (2007) detected that at higher turbulence intensities, the leading-edge bluntness plays an important role. Goldstein and Sescu (2008) further investigated this issue and showed that unsteady disturbances with small amplitudes were amplified in the presence of a blunt
leading edge. Further downstream, this base flow becomes inflectional and inviscidly unstable. Wave-packet like disturbances can grow rapidly on these inflectional profiles.

Today, it is known that the leading edge is the key receptivity site for the penetration of free-stream disturbances as it has the thinnest boundary layer. An earlier experiment by Fransson et al. (2005) in the presence of FST with intensities between 1.4 %

and 6.7 % over a flat plate showed that the initial energy in the boundary layer is proportional to the square of the turbulence intensity. Furthermore, the energy grows linearly with the Reynolds number based on the downstream distance.

A study on the flow through a compressor passage by Zaki et al. (2010) showed that at moderate FST of up to 3 %, upstream of separation the amplification of Klebanoff distortions is suppressed in the favorable pressure gradient region (FPG). Thus, the FPG is normally stabilizing the flow with respect to bypass transition. The instantaneous separated shear region, however, included Klebanoff distortions. At an increased $TI$ of approximately 6.5 % they found turbulent patches that cause local attachment but the spanwise averaged data showed no attached flow. The percentage of time, where the spanwise averaged flow on the suction surface is attached, was found to be 59.8 % at a $TI$ of 8 % and 96.6 % at a $TI$ of 10 % which indicates that there were instances of separation even at a high $TI$ of 10 %.

In case of bypass transition, shear sheltering is a mechanism that relates free-stream disturbances to the generation of streaks within the boundary layer via non-modal growth (Butler and Farrell, 1992). The phenomenon of shear sheltering (Hunt and Carruthers, 1990) permits disturbances to penetrate up to a certain depth into the boundary layer. Jacobs and Durbin (1998) showed that the penetration depth of a disturbance depends on the frequency and Reynolds number with lower frequencies penetrating deeper into the boundary layer. Zaki and Saha (2009) complemented their analytical solutions with a physical interpretation of shear sheltering which contrasts low and high frequency modes and their ability to penetrate the shear layer. Zaki (2013) further illustrated this filtering effect of the boundary layer using a model problem with two time scales.

Streak instabilities can be visualized through the meandering of the streaks. Both sinusoidal and varicose modes have been identified (Asai et al., 2002; Mandal et al., 2010). In contrast to prior research that considered idealized streaks obtained as a boundary layer response to well-defined forcing, for example a streamwise vortex (Andersson et al., 2001) or by using a single inflow continuous Orr-Sommerfeld mode (Vaughan and Zaki, 2011) where the streaks were periodic in the spanwise direction. Hack and Zaki (2014) studied the response of the boundary layer beneath FST. They observed streaks of different amplitudes, sizes and orientations with secondary instabilities being sporadic and localized on particular streaks. The instability mode was found to be either the outer or inner type (Vaughan and Zaki, 2011) and the prevalence depends on the parameters of the flow with the outer mode being dominant in zero-pressure-gradient boundary layers (Jacobs and Durbin, 2001). This issue was confirmed by Brandt et al. (2004) with the sinusoidal scenario more likely to occur. Zaki and Durbin (2005) showed that only two free-stream modes are sufficient for the complete transition process. These are a low-frequency component that penetrates the shear layer causing the formation of streaks and a high-frequency component that does not sufficiently penetrate the boundary layer due to shear sheltering but provides excitation for the growth of the outer instability.

In a study using steady base-streaks, Andersson et al. (2001) computed the outer instability using an inviscid secondary instability theory and found that the critical streak amplitude is about 26 % of the free-stream velocity for the sinusoidal instability mode and about 37 % for the more stable varicose mode. Vaughan and Zaki (2011) relaxed the steady base-flow assumption and found that when the base streaks are unsteady, the sinusoidal and varicose modes merge. Furthermore, they observed that the outer mode of instability emerged at a critical streak amplitude of 8.5 %. A common scenario of the inner instability mode is a local overlap of the leading edge of a high-speed streak with that of the trailing edge of a low-speed streak causing the local velocity profile to become inflectional. This leads to a mainly varicose instability in the vicinity of the wall.

For example, in the presence of a blunt leading edge, Nagarajan et al. (2007) found evidence of an inner instability mode as this mode with a relatively low phase speed is more effectively excited due to receptivity at the leading edge (Vaughan and Zaki, 2011). Using a stability analysis, Marquillie et al. (2011) showed that with increasing adverse pressure gradient the secondary instability changed from the sinusoidal to the varicose mode.

To overcome the well-known drawbacks of DNS, wall-resolved large-eddy simulations (LES) can be used. In the LES of bypass transition under high FST by Yang et al. (1994) the constant coefficient subgrid-scale (SGS) model had to be modified in an *ad hoc* manner to reduce the dissipation before transition. The inadequacy of constant coefficient SGS models is also shown in Sayadi and Moin (2011). To alleviate some of these drawbacks, Germano et al. (1991) proposed a dynamic SGS model, where the subgrid-scale stresses vanish in laminar flows and at solid boundaries guaranteeing the correct asymptotic behavior in the near-wall region. The results of LES of transitional and turbulent channel flow based on this dynamic SGS model showed good agreement with DNS. Sayadi and Moin (2011) compared different SGS models based on the predicted skin friction coefficient along a flat plate and showed that dynamic SGS models are capable of predicting the point of transition accurately and independently of the transition scenario.

Lardeau et al. (2012) conducted a LES to predict transitional separation bubbles. Their main objective was to compare the results with a DNS by Wissink and Rodi (2006) with emphasis on the response to FST. The essential features of the transition process could be captured at a resolution of around 10 % of the equivalent DNS with an appropriate SGS model. Based on LES Rao et al. (2014) showed that the mixing caused by Klebanoff modes due to FST is unsteady while that due to roughness is steady. The net effect of mixing was a shift in the inflection point of the velocity profile towards the wall, thus promoting earlier transition. The spanwise waviness of the Kelvin-Helmholtz vortex and its eventual breakdown to turbulence was also associated with the unsteady mixing in the presence of FST.

Diwan and Ramesh (2009) performed an experimental and theoretical study of a laminar separation bubble and its associated linear stability mechanisms on a flat plate set-up in a wind tunnel such that there was an imposed pressure gradient typical for an airfoil with separation. They observed an exponential growth rate of disturbances in the region upstream of the mean maximum height of the bubble which is indicative for a linear stability mechanism. They further find that the primary instability mechanism in a separation bubble is inflectional in nature and its origin can be traced back upstream of the separation region. The key conclusion of their study is that the inviscid inflectional instability of the separated shear layer should be seen as an extension of the instability of the upstream attached adverse pressure gradient boundary layer. Furthermore, only when the separated shear layer has moved considerably away from the wall, a description by the Kelvin-Helmholtz instability paradigm with its associated scaling principles becomes relevant.

The Brite-Euram project LESFOIL (Mellen et al., 2003) focused on assessing the feasibility of LES for the computation of the flow around an airfoil at a Reynolds number of two million. It was found that on a very fine mesh, the resolution was sufficient to capture the transition process without numerical forcing. A good agreement with experiments was observed. Breuer (2018) and Breuer and Schmidt (2019) investigated the effect of artificially generated isotropic inflow turbulence on the flow around an airfoil using wall-resolved LES on a fine grid. A perceptible influence of the turbulence intensity as well as the time and length scales of the inflow turbulence on the development of the flow field around the airfoil was found.

To better understand the process of laminar to turbulent transition on wind turbines, experimental studies have been conducted. Within the MexNext project (Boorsma et al., 2018) first comparisons between measurements and simulations were carried out for a rotor model in a wind tunnel. Furthermore, transition was also successfully detected in the experimental data (Lobo et al., 2018), where the growth of T-S disturbances was identified as the main mechanism for the transition process. Also as a part of the MexNext project, seven transitional CFD computations from four groups were carried out and reasonable agreement between the transition location determined based on the experimental data and the simulations was found (Schaffarczyk et al., 2018). In Schaffarczyk et al. (2017) measurements on a rotor blade were carried out in the free atmosphere to study the behavior of the boundary layer within a specific zone on the suction side at different operational states. Laminar and turbulent flow was distinguished. It was found that in case of atmospheric flow, the turbulent energy spectrum possesses a maximum at about 0.01 Hz with a decrease according to Kolmogorov's $k^{-5/3}$ law. However, in wind tunnels with a turbulence grid much more energy is distributed in the kHz range. Thus, it became clear that the concept of Mack (1977) to correlate the $TI$ to the $N$-factor may be questioned in case of atmospheric inflow without further assumptions about a low-frequency cut-off.

In Reichstein et al. (2019) microphone and pressure sensor measurements together with thermographic imaging to study transition on a blade of 45 m length were collected. A good agreement between both the data of the microphones and the thermographic imaging technique was found. The laminar-turbulent transition location in an associated RANS simulation of the transitional flow, wherein an $N$-factor with respect to Mack's correlation and corresponding to the inflow turbulence intensity was set-up, deviated from the experimental results. Consequently, it was proposed to conduct wall-resolved LES to better understand the transition process within atmospheric turbulence.

According to this long-term goal of running wall-resolved LES with modeled atmospheric inflow turbulence of appropriate length and time scales for a better understanding of the transition modes occurring at Re numbers of several millions, it was decided to step up Re incrementally. The reason is that transitional studies using wall-resolved LES around airfoils for Re numbers in the order of a few million are rarely available. Hence, the present study comprises wall-resolved LES with modeled isotropic atmospheric inflow turbulence carried out at a chord Reynolds number of 100,000 for a 20 % thickness airfoil corresponding to the test section of the aforementioned experiment (Reichstein et al., 2019). This paper is intended to contribute to the growing knowledge on transition mechanisms, in particular bypass transition and to investigate the mode of transition as a function of atmospheric inflow turbulence intensity. An area of particular interest was to look at how much energy of the low-frequency disturbances penetrates into the boundary layer in comparison to a case without any added inflow turbulence.

The paper is organized as follows: Section 2 briefly summaries the main features of the underlying simulation methodology. The description of the flow cases including the numerical setup (grid, boundary conditions, inflow turbulence) is provided in Section 3. The results are evaluated and discussed in Section 4 and conclusions are drawn in Section 5.

## 2 Simulation approach and numerical method

The simulation methodology relies on a classical wall-resolved large-eddy simulation extended by an inflow generator as explained below. The main features are summarized in Table 1. The filtered three-dimensional, time-dependent Navier-Stokes equations for an incompressible fluid are solved based on a finite-volume method on block-structured grids, which is second-order accurate in space and time (Breuer, 1998, 2000, 2002). The additional subgrid-scale stress tensor mimics the influence of the non-resolved small-scale structures on the resolved large eddies. In the present study, the widely used dynamic version of the classical Smagorinsky (1963) model is applied which was introduced by Germano et al. (1991) and Lilly (1992). As mentioned in the introduction, the dynamic variant has several advantages compared to the constant coefficient SGS model and is a must for the prediction of transitional flows. As detailed in Section 3, the near-wall grid resolution is fine enough to resolve the viscous sublayer. Thus, Stokes no-slip condition is applied at the surface of the airfoil.

Synthetic turbulence inflow generators (STIG) based on a variety of different techniques were suggested in the literature, see, e.g., the review by Tabor and Baba-Ahmadi (2010) evaluating the pros and cons of the different techniques. In the present study the digital filter method originally proposed by Klein et al. (2003) is applied to generate artificial turbulent inflow data. Presently, the more efficient procedure suggested by Kempf et al. (2012) is used, since it reduces the computational effort and memory requirements significantly compared to the original method by Klein et al. (2003). The method relies on discrete linear digital non-recursive filters which depend on certain statistical properties to be defined by the user. These are profiles of the mean velocity and Reynolds stresses and the definition of one integral time scale ($T$) and two integral length scales ($L_y$, $L_z$). These quantities are sufficient to generate artificial turbulence with proper autocorrelations in time and two-point correlations in space. For this purpose, the filter coefficients are multiplied with a series of random numbers characterized by a zero mean and a unit variance. Thereby, the filter coefficients describe the two-point correlations and the autocorrelation of the inflow turbulence. A required three-dimensional correlation between the filter coefficients is achieved by the convolution of three one-dimensional filter coefficients. The cross-correlations between all three velocity components and thus the representation of a realistic inflow turbulence is guaranteed by the application of the transformation by Lund et al. (1998).

In the present setup the inflow turbulence is not imposed at the inlet of the computational domain but within the domain using a special source-term formulation developed and validated in De Nayer et al. (2018), Schmidt and Breuer (2017), Breuer (2018) and Breuer and Schmidt (2019). The idea behind the source-term formulation is that it enables the injection of inflow turbulence in sufficiently resolved flow regions. This measure prohibits that it is damped out by numerical dissipation before reaching the region of interest. For external flows as considered in the present study, the inlet region is not resolved by a fine grid, which leads to a strong damping up to a complete cancellation of small flow structures. However, the regions of main interest are the boundary layers and the wake region, where the grid is strongly clustered and thus is sufficiently fine to resolve these structures. Consequently, the injection of the inflow turbulence is done in this region as detailed in Section 3. The artificial velocity fluctuations generated by the digital filter method explained above are introduced as source terms directly into the momentum equation. In Schmidt and Breuer (2017) the method was shown to guarantee the correct distribution of the autocorrelations. Besides the application to channel flows (Schmidt and Breuer, 2017) the source term methodology was

also successfully applied to the bluff-body flow past a wall-mounted hemisphere (De Nayer et al., 2018) and the flow around the SD7003 airfoil (Schmidt and Breuer, 2017; Breuer, 2018; Breuer and Schmidt, 2019). For more information about the validation of the method, we refer to Schmidt and Breuer (2017), Breuer (2018), Breuer and Schmidt (2019) and De Nayer et al. (2018).

**Table 1.** Finite-volume method and models used for LES.

| Property | Feature |
|---|---|
| **fluid** | incompressible |
| **grid type** | curvilinear, block-structured |
| **variable arrangement** | cell-centered, non-staggered |
| **discretization of integrals** | midpoint rule |
| **interpolation scheme** | linear interpolation |
| **accuracy in space** | second-order accurate |
| **solution scheme** | predictor-corrector time-marching scheme |
| **predictor** | low-storage Runge-Kutta scheme |
| **corrector** | pressure-correction method |
| **accuracy in time** | second-order accurate |
| **pressure-velocity coupling** | momentum interpolation technique (Rhie and Chow, 1983) |
| **turbulence modeling** | large-eddy simulation (Breuer, 1998, 2000, 2002) |
| **SGS model** | dynamic Smagorinsky model (Germano et al., 1991; Lilly, 1992) |
| **wall treatment** | wall-resolved LES (Stokes no-slip condition) |
| **inflow turbulence** | digital filter concept (Klein et al., 2003; Kempf et al., 2012) |
| **inflow injection** | source-term formulation (De Nayer et al., 2018; Schmidt and Breuer, 2017; Breuer, 2018; Breuer and Schmidt, 2019) |

## 3   Description of the flow case and numerical setup

### 3.1   Description of the flow case

The airfoil profile with a relative thickness of 20 % used for this study corresponds to the profile at a radius of 35 m on a wind turbine blade of the type LM45.3p as used on the 2 MW Senvion (formerly REpower) MM92 wind turbine. This corresponds to the test section of the experiment (Reichstein et al., 2019) of interest. The study focuses on the flow past this airfoil at an angle of attack of $\alpha = 4°$, which lies within the range measured during the experiment. A Reynolds number of $\mathrm{Re}_c = 10^5$ based on the free-stream velocity $u_\infty$ and the chord length $c$ was chosen for the current study. Note that all results presented below are non-dimensionalized by these two parameters. The current case is thought as a build up to the higher Re numbers of the

experiment which lie in the order of a few million as described in Section 1. A parameter study with varying inflow turbulence intensities of $TI = 0$, 1.4 %, 2.8 %, 5.6 % and 11.2 % was carried out.

## 3.2 Numerical setup

### 3.2.1 Computational domain and grid resolution

The numerical simulations used a C-type grid with the angle of attack already included in the base mesh. The mesh extends eight chord lengths upstream of the leading-edge of the airfoil and fifteen chord lengths downstream of the trailing-edge to avoid the influence of the outflow boundary condition on the flow around the airfoil. This is a standard domain size which is sufficient for LES as also seen in, for e.g. Mellen et al. (2003); Ke and Edwards (2017); Gao et al. (2019); Solís-Gallego et al. (2020). A suitable choice of the spanwise extension of the computational domain is critical for such geometrically two-dimensional airfoils. The spanwise width is set to $z/c = 0.25$ as seen in Schmidt and Breuer (2017) and Breuer (2018) for a Reynolds number of $60,000$. At this Reynolds number, Galbraith (2009) found marginal differences for cases with spanwise widths between $z/c = 0.1$ and $0.3$. Furthermore, Schmidt (2016) has carried out simulations with $z/c = 0.25$ and $0.5$, where two-point correlations were evaluated. A width of $z/c = 0.25$ was found to be sufficient. Transitional studies in the LESFOIL project (Mellen et al., 2003) at $\mathrm{Re}_c$ of $2 \times 10^6$ used a spanwise extension of $z/c = 0.012$ which corresponds to a spanwise width scaled down by a factor of 20 in comparison to that of the present study while the flow has a Reynolds number scaled down by a factor of 20 compared to that of the LESFOIL project.

The quality of the grid was given high importance with special focus on grid orthogonality, low expansion factors and the maintenance of grid smoothness within the computational domain. It satisfies the grid resolution requirements for a wall-resolved LES as outlined by Piomelli and Chasnov (1996) with a $y_{1st}^+ < 2$ for the wall-normal resolution, $\Delta x^+ = \mathcal{O}(50 - 150)$ for the streamwise resolution and $\Delta z^+ = \mathcal{O}(15 - 40)$ for the spanwise resolution. Furthermore, the grid resolution applied satisfies the requirements proposed by Asada and Kawai (2018) who performed a grid convergence study for wall-resolved LES including separation bubbles and found grid-independent results for $c_p$ and $c_f$ as well as a sufficient resolution for resolving streaks using a grid with $\Delta x^+ = \mathcal{O}(25 - 50)$ for the streamwise, $\Delta z^+ = \mathcal{O}(13 - 30)$ for the spanwise and $y_{1st}^+ = 0.8$ in the wall-normal direction, respectively.

The computational domain is discretized by 56 control volumes (CVs) ($\Delta z^+ \leq 25$) in the spanwise direction. In the tangential direction it is discretized by 240 CVs ($\Delta x^+ \leq 30$) on the suction side of the airfoil and by 162 CVs ($\Delta x^+ \leq 60$) on the pressure side. Since the transition process on the suction side is of special interest, a finer grid resolution is applied there. To allow for the resolution of the viscous sublayer, the resolution near the wall satisfies the condition $y_{1st}^+ < 1$ (first cell center). With a mild expansion factor of the geometrical series of 1.05 for the wall-normal direction, there are 193 CVs in the wall-normal direction and 169 CVs in the wake region. Overall, the curvilinear block-structured grid consists of about $8.2 \times 10^6$ CVs and 73 blocks. Parallelization is achieved based on grid partitioning with the classical domain decomposition and MPI.

The dimensionless simulation time step is set to $\Delta t \, u_\infty / c = 5 \times 10^{-6}$ which corresponds to a low maximum CFL number of about 0.05. The simulations were run for a total of $3.34 \times 10^6$ time steps with $1.7 \times 10^6$ averaging time steps covering 8.5

dimensionless time units. Each simulation was run in parallel on 73 processors consuming approximately $80.5k$ CPU-hours each.

Since the analysis is solely based on numerical simulations, it was necessary to run a grid convergence study to show that the grid used is sufficient in terms of its resolution for the objectives of the current investigation. The computational setup described above (standard grid) is the one used for the present study. The grid independence study was conducted by applying
a refined grid with about three times more grid points. A comparison of the most important grid parameters is given in Table 2. For the purpose of this study, it is crucial that the transition processes predicted on both grids are the same. Fig. 1 shows a comparison using the dimensionless Q-criterion. From Figs. 1a and 1b it is clear that when using both the standard and the refined grid spanwise rolls are observed upstream of the separation with larger rolls associated with a Kelvin-Helmholtz (K-H) type of instability seen downstream of the separation which finally breaks down to turbulence. In both cases, the location of the
laminar separation and turbulent reattachment as well as the general transition process and location of the K-H rolls are very similar.

Clearly visible, high frequency streamwise components appear around the mid-blade especially seen in Fig. 1a, which have to be explained. These are caused by numerical noise and are only visible near the region of breakdown to turbulence and according to our analysis do not directly affect the transition process. The cause was found to be some minor numerical
oscillations due to the application of the central second-order accurate scheme. This scheme has the advantage of low numerical dissipation, which is important for LES and especially the simulation of transitional flows. On the other hand, it is prone to numerical oscillations. To show that the numerical scheme is indeed the cause, the case with $TI = 0$ % was separately run using a blended scheme, that is, a blend between a standard 98 % central difference scheme (CDS) and a 2 % standard upwind scheme. The blended scheme is hereafter referred to as 98 % CDS. The resulting Q-criterion plot is depicted in Fig. 1c. By
comparing with Fig. 1a it is clear that the high frequency streamwise components have noticeably reduced when the blending scheme is used. However, the application of the blended scheme with a 2 % upwind contribution did not alter the transition process and only a slight change to the separation region is visible on the $c_p$ and $c_f$ distributions depicted in Fig. 2a. Since there are no changes to the transition scenario, it was not deemed necessary to apply the blended scheme to the other cases with added inflow turbulence and restart the entire simulations as this would consume the limited computational resources needed
for the simulations at higher Re. Thus, all results discussed hereafter and especially in Section 4 refer to the use of the standard central second-order accurate scheme (100 % CDS) unless explicitly stated otherwise.

Figure 2a also shows a comparison of the pressure coefficient between the standard and the refined grid. For comparison purpose, data from XFOIL (Drela, 1989) are also included for a $N_{crit}$ value of 9, where $N_{crit}$ is the log of the amplification factor of the most amplified frequency that triggers transition. The corresponding $c_f$ plot can be seen in Fig. 2b. Slight deviations
between the results on both grids are visible. However, for the current study which is focused on the transition phenomena, the standard grid provides a sufficiently accurate resolution with no significant changes observed in the mode of transition as discussed above. Additionally, the suction side is of special interest in our study, which possesses a finer grid resolution than the pressure side. Details are found in Table 2. Taking into account the goal of the present study and the very high computational

**Table 2.** Parameters for the grid convergence study.

| Grid Parameters | Standard Grid | Refined Grid |
|---|---|---|
| $y_{1st}^+$ (first cell center) | $< 1.0$ | $< 0.5$ |
| $\Delta x^+$ (suction) | $\leq 30$ | $\leq 15$ |
| $\Delta x^+$ (pressure) | $\leq 60$ | $\leq 30$ |
| $\Delta z^+$ | $\leq 25$ | $\leq 15$ |

costs already necessary for the long-lasting time-consuming predictions on the standard grid, the resolution of the standard grid is deemed to be sufficient for the purpose of this study.

Furthermore, it is unexpected to see deviations mainly in the laminar region of the flow on the suction side hinting at another possible reason not directly related to grid independence. The peak in the $c_f$ plot at around 10 % chord in the case of the refined grid is similar to what is seen in preliminary studies for a Reynolds number of 500k of the same airfoil. This deviation in the friction coefficient is caused by the airfoil geometry not being sufficiently smooth, an issue that becomes increasingly prominent with increasing grid resolution. By fixing the airfoil smoothing issue, the case at Re = 500k experiences an increase in the favorable pressure gradient and a smoothening of the $c_f$ curve. It is very likely that the same issue is at play on the refined grid at Re = 100k. From the plot of the displacement and momentum thickness (Fig. 5e and 5f) it is obvious that the boundary layer properties in the laminar region converge quite well, further indicating that the $c_p$ and $c_f$ distributions found on the blade surface arise due to airfoil smoothing issues.

Figure 5c shows the shape factor which is the ratio of the displacement thickness to the momentum thickness. The results on the standard grid with 98 % CDS and the refined grid agree quite well, but a clear discrepancy between the predicted data on the standard grid at 100 % CDS and 98 % CDS is visible. This is a result of the amplification of small variations in the displacement and momentum thicknesses on account of the way in which the shape factor is calculated. However, the location of the separation bubble (see Table 3) and the corresponding location of transition onset indicated by the peak in the shape factor match quite well for these cases. As discussed in Asada and Kawai (2018), the grid resolution of the standard grid on the suction side is sufficient for the study of transition including separation bubbles, but a finer grid resolution could better capture the vortex development. This explains the difference in the displacement and momentum thickness as well as the shape factor between the standard and the refined grid within the region of the separation bubble. This does not, however, affect the mode of transition.

### 3.2.2 Boundary conditions

At the inlet plane that encompasses the entire circumference of the "C", individual velocity components are specified with a streamwise velocity $u/u_\infty = 1$ while the other components are set as $v_\infty = w_\infty = 0$. No perturbations are added on the inlet plane, therefore, a zero turbulence level is assumed. This is done because even with the addition of turbulence of appropriate characteristics, the probability of the high frequency components reaching the airfoil would be small due to the lower grid

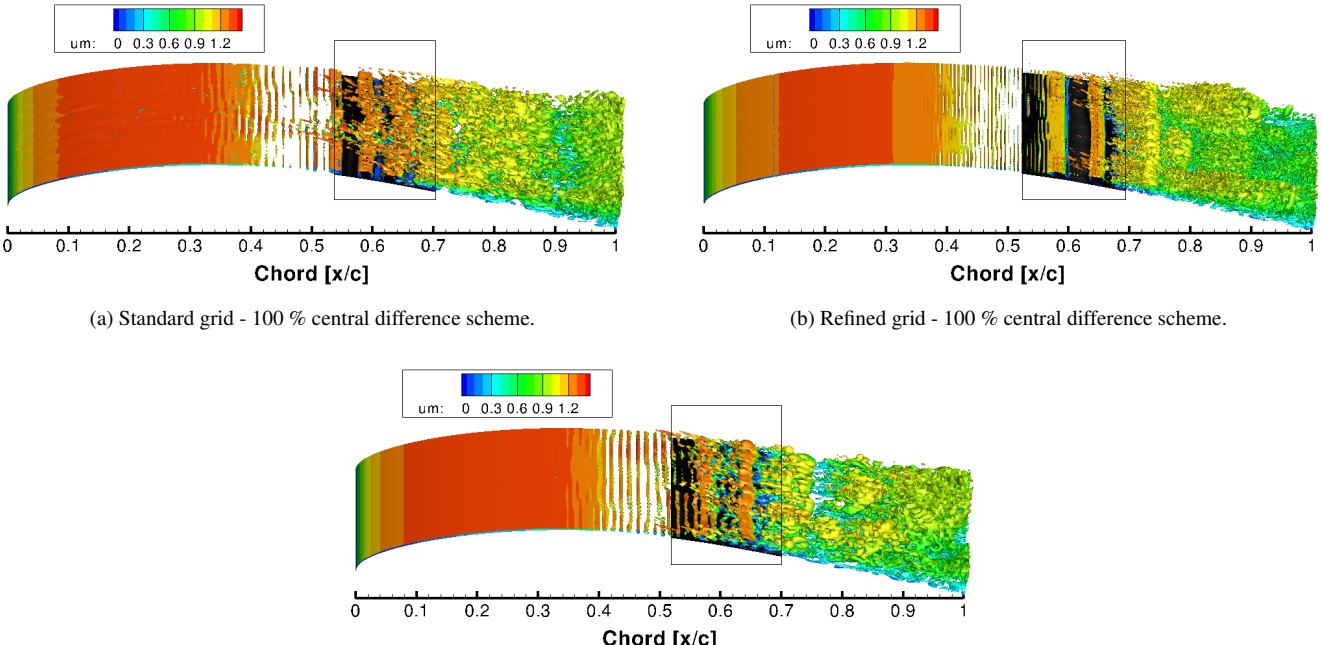

(a) Standard grid - 100 % central difference scheme.

(b) Refined grid - 100 % central difference scheme.

(c) Standard grid - blending with 98 % central difference and a 2 % upwind scheme.

**Figure 1.** Instantaneous iso-surfaces of the $Q$-criterion ($Q = 250$) colored by the mean streamwise velocity normalized by $u_\infty^2/c^2$. The $Q$-criterion is commonly used for vortex visualization and defines vortices as areas where the vorticity magnitude is greater than the magnitude of the rate of strain (Hunt et al., 1988). The black rectangle highlights the mean separation region which is visualized below by the black iso-surface ($u_{mean}$ = -0.01) indicating negative mean velocity.

resolution in the vicinity of the inlet boundary. Instead, in the cases where inflow turbulence is necessary, it is injected within the computational domain as described in Section 3.2.3.

On the outlet plane that encompasses the open end of the "C", a convective boundary condition is set as this ensures that vortices can pass the boundary without significant disturbances or reflections back into the inner domain. The boundary conditions reads $\partial u_i/\partial t + u_{\text{conv}}\, \partial u_i/\partial x = 0$, where $u_{\text{conv}}$ is the mean convection velocity set to $u_\infty$. Periodic boundary conditions are

applied in the spanwise direction as the effective number of unknowns is smaller, thereby reducing computational costs while also providing the advantage that it is possible to simulate small sections that are not terminated by a surface. Stokes no-slip condition is applied at the surface of the airfoil.

### 3.2.3 Injection of inflow turbulence

As described in Section 2, the generation of inflow turbulence data requires an appropriate profile of the Reynolds stresses,

the definition of one integral time scale $T$ and two integral length scales in the lateral and spanwise directions, $L_y$ and $L_z$,

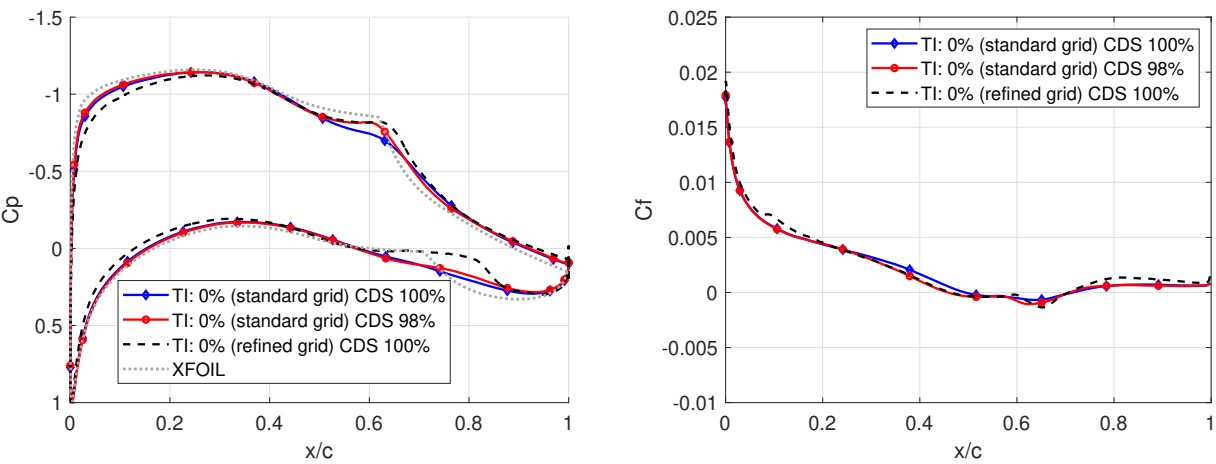

(a) Pressure coefficient of the standard vs. the refined grid. Here, the XFOIL results by Drela (1989) ($N_{crit} = 9$) are also included.

(b) Suction side friction coefficient of the standard vs. the refined grid.

**Figure 2.** Comparison of time and spanwise averaged statistics at $TI = 0$ %.

respectively. Note that the integral time scale can also be interpreted as a measure for an integral length scale in streamwise direction as explained below. The present study relies on the simplifying assumption that the approaching flow is isotropic. This has been shown to approximately hold true, for instance, after a sufficiently long distance behind passive grids in wind tunnels (Herbst et al., 2017; Mohamed and Larue, 1990; Roach, 1987).

In case of isotropic turbulence, the Reynolds shear stresses are set to zero and the three normal Reynolds stresses are equal $\overline{u'u'} = \overline{v'v'} = \overline{w'w'}$ and have a constant value across the entire turbulence inflow plane. This value depends on the chosen turbulence intensity $TI = \sqrt{\overline{u'u'}}/u_\infty$. Five different turbulence intensities are studied as outlined in Section 3.1. The case without inflow turbulence ($TI = 0$ %) is taken as the reference. Next, as outlined in Section 1 experiments have suggested that at a $TI > 1$ % the transition mechanism is known to deviate from the typical T-S route. Therefore, a slightly higher value of

$TI = 1.4$ % has been chosen, followed by a series in which the turbulence intensity is doubled up to a $TI = 11.2$ %, as in the study by Breuer (2018).

The integral time scale and the two integral length scales are chosen based on the experimental data of Hain et al. (2009) since the Reynolds number of the experiment is of similar order as that studied here. The length scales are easily determined from the autocorrelation function of the experimental data, for example using the integral over the autocorrelation function

to the first zero crossing to determine the time lag. Since the autocorrelation function is usually an exponential function, it is common practice to determine the time scale as the time lag at which the autocorrelation function has decreased to 1/e or 0.37 (Foken, 2008). In dimensionless form, the integral time scale is given by $T \cdot u_\infty/c = 0.118$. According to Taylor's hypothesis of frozen turbulence the two dimensionless integral length scales are $L_y/c$ and $L_z/c = 0.118$. Again, the same length and time scales were previously applied in Breuer (2018) and Breuer and Schmidt (2019).

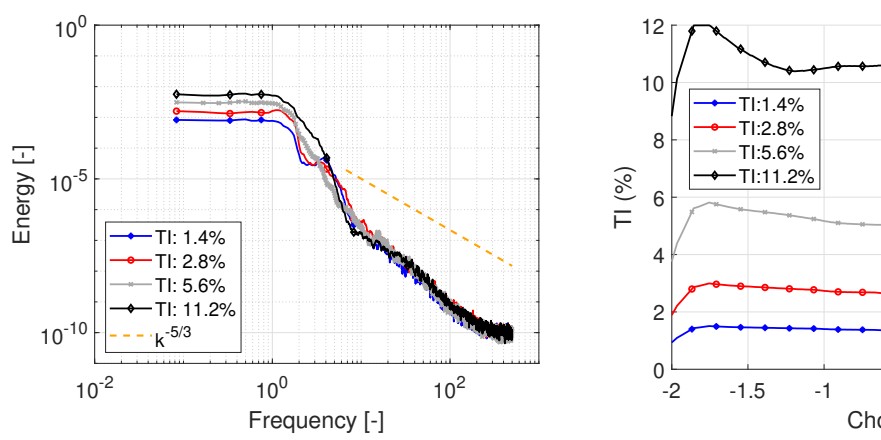

(a) Turbulent kinetic energy spectra of generated inflow turbulence.

(b) Turbulence intensity development with downstream distance measured at a constant wall-normal distance corresponding to the boundary layer thickness at 50 % chord.

**Figure 3.** Inflow turbulence characteristics.

Another reason for using data from the experiment by Hain et al. (2009) is the fact that the scales from the experimental data (Reichstein et al., 2019) are very large and would require a large computational domain also in the spanwise direction (up to 8 times the chosen spanwise extension) which would make the task computationally infeasible. For future simulations at higher Reynolds numbers as discussed earlier, anisotropic inflow turbulence will be generated based on the Kaimal formulation (IEC61400-1) wherein the length and time scales are relatively determined based on a single integral scale. In this case, the

scales will be determined based on the spanwise extent of the domain and not on the experimental data, again due to the computational costs being a limiting factor.

Figure 3a shows the turbulent kinetic energy spectra of the generated inflow turbulence. The resulting turbulence possesses a maximum at the lower end of the frequency spectra, similar to that of atmospheric turbulence, as described in Section 1 and seen in Schaffarczyk et al. (2017). Fig. 3b shows the downstream development of the free-stream turbulence. The decay

of free-stream turbulence is plotted using the averaged Reynolds stresses at the end of the simulation period defined as $TI = \sqrt{\frac{1}{3} \times (\overline{u'u'} + \overline{v'v'} + \overline{w'w'})}/U$, where $\sigma_1^2 = \overline{u'u'}$ etc. are the averaged normal Reynolds stresses in the three principle directions and $U$ denotes the mean inflow velocity.

The peak seen at about $x/c$ = -2 (close to injection plane) is due to the way in which the turbulence is injected into the domain as outlined in Section 2. The injection area is located about two chord lengths upstream of the leading-edge ($L_s/c = 2$)

as seen in Fig. 4. Schmidt (2016) has shown that this is a sufficient distance from the area of interest to ensure the correct development of the injected turbulence inflow data. The streamwise extension of the $STIG$ injection region is set to twice the integral length scale $L_x$ which can be determined from the integral time scale and Taylor's hypothesis as $L_x = u_\infty \cdot T$.

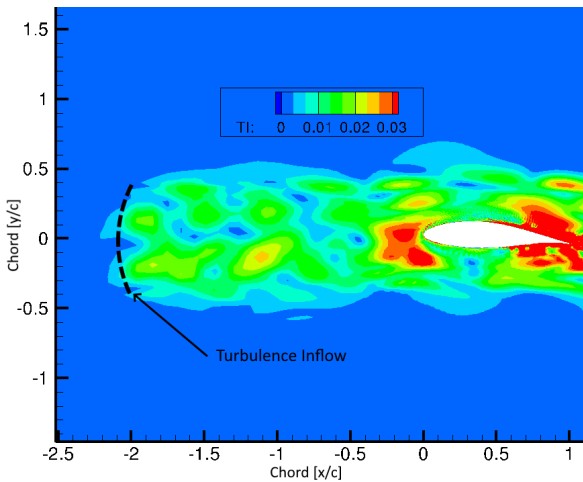

**Figure 4.** Instantaneous spatial distribution of turbulent fluctuations with an inflow $TI$ of 1.4 %.

Furthermore, the injection area measures 0.8 chord lengths in the cross-stream direction and covers the entire domain in the spanwise direction.

The required turbulence intensity is expected to be achieved slightly upstream of the airfoil. An analysis of the development of the inflow turbulence was conducted by Breuer (2018) for the same inflow turbulence conditions. A development length of about one chord length was found to be sufficient depending on the turbulent length scale. In the present case, the fully developed turbulence is seen at about $x/c$ = -1 to -0.2, which is just upstream of the region influenced by the airfoil.

In all cases, the airfoil seems to influence the development of $TI$ beginning at about $x/c$ = -0.2. An increase in effective $TI$ up to the separation/transition point (around 50 % chord) is observed before it begins to drop. In the case of $TI = 11.2$ % there is no separation bubble and this seems to be the reason why there is a continuous decay in turbulence (after the small increase above the leading edge of the airfoil as also seen in the other cases). It probably has something to do with the boundary layer being thinner in this case due to the absence of a separation bubble and the fact that the analysis for the calculation of $TI$ is conducted at a height corresponding to the boundary layer thickness at 50 % chord for the $TI = 0$ % case.

## 4   Results

The results from the simulations using the standard grid and the pure CDS scheme (100 %) are presented here as follows. Some plots may include data from the case with the blended scheme (CDS of 98 %), but need no additional discussion other than that already discussed in Section 3.2.1 with respect to these plots. First, in Section 4.1 a comparison is made between different aerodynamic properties of the airfoil at varying inflow turbulence intensities. This is followed by a study on the growth of streamwise velocity disturbances which helps to understand the underlying mode of transition (Section 4.2). Furthermore, a visual representation of the transition phenomena is provided. In Section 4.3 power spectral density distributions are evaluated

to further investigate the type of transition as a function of the inflow turbulence intensity. Finally, boundary layer streaks are analyzed in Section 4.4. Here it should also be clarified that no direct comparisons with the experiment by Reichstein et al. (2019) are made since the present simulations are carried out at a Reynolds number of 100,000 whereas the experiment is conducted at a Reynolds number in the order of a few millions. Nevertheless, the airfoil considered and the angle of attack corresponds to this experiment for the purpose of future comparisons when higher Reynolds numbers with anisotropic inflow turbulence are considered as discussed earlier.

## 4.1 Influence of turbulence intensity on aerodynamic properties

Figure 5 depicts different characteristic properties of the airfoil, i.e., the pressure coefficient $c_p = (p - p_\infty)/(0.5\,\rho\,u_\infty^2)$, the friction coefficient $c_f = \tau_w/(0.5\,\rho\,u_\infty^2)$, the shape factor $H_{12}$ and the lift-to-drag ratio $c_L/c_D$, as a function of the inflow turbulence intensity. All these quantities are based on the averaged flow, i.e., averaging in time and in spanwise direction. The separation and reattachment points in Table 3 are determined from the distribution of $c_f$. The shape factor $H_{12}$ is defined as the ratio between the displacement thickness $\delta_1$ and the momentum thickness $\delta_2$, both being integrated in the wall-normal direction from the surface of the airfoil to the location where the local dimensionless edge velocity $U_e(x)/U_{max}(x) = 0.99$ is reached.

The distribution of the friction coefficient of the simulations at different turbulent inflow conditions shows an expected, albeit small downstream shift of the separation point with increasing $TI$, up to a $TI$ of 5.6 % as seen in Table 3. At $TI = 11.2\,\%$ the separation bubble vanishes. Tangermann and Klein (2020) have studied the influence of inflow turbulence with varying intensities and length scales on the NACA0018 airfoil, which has a similar thickness as the airfoil studied here. In their case the separation bubble already originates between 25 and 35% chord, which is further upstream than that seen in the current simulations. This can be attributed to the location of the maximum thickness of the airfoil since the adverse pressure gradient favors separation. In case of the NACA0018, its maximum thickness is located at 30 % chord, while the airfoil studied here has its maximum thickness located at 36 % chord.

Downstream of the separation point, a nearly constant distribution of $c_f$ is found for approximately 10 % chord length beyond which the crucial transition onset is recognized through the development of a minimum of $c_f$ which leads to a turbulent reattachment further downstream. For example, for the $TI = 0\,\%$ case, the crucial transition onset lies somewhere in-between the region where $c_f$ begins to drop (around 58 % chord length) and the point of minimum $c_f$ (around 64 % chord length). In the presence of a separation bubble, the transition process occurs within the bubble, i.e., between 50 and 75 % chord. At a $TI$ of 11.2 % the flow turns fully turbulent around 60 to 70 % chord as indicated by the maximum slope of the friction coefficient.

The crucial transition onset can also be defined as the location where the maximum value of the shape factor is found on the suction side of the airfoil, similar to the investigations of Burgmann and Schröder (2008). A good correlation with the previous definition that uses $c_f$ is seen. In the presence of added inflow turbulence, the maximum value of the shape factor is shifted upstream with increasing $TI$ as a consequence of the earlier transition onset discussed below. In all cases, a shape factor of about 2.2 is found downstream of the turbulent reattachment. This is a typical value for the shape factor of turbulent flows near separation (Castillo et al., 2004).

The area under the curve of the pressure coefficient decreases with rising $TI$, although the effect is marginal in some cases. On the other hand, with an increase in $TI$, a corresponding increase in the friction coefficient within the region upstream of transition is observed in Fig. 5b. The effect of these variations in $c_p$ and $c_f$ on the airfoil performance can be seen in the lift-to-drag ratio depicted in Fig. 5d. Thus, the performance of the airfoil characterized by its lift-to-drag ratio decreases with increasing $TI$, whereas the opposite trend is seen in the study by Breuer (2018). A short discussion on the reason for this disparity follows.

On the relatively thinner airfoil (8.51 %) in the study by Breuer (2018), separation takes place close to the leading edge at around 20 % chord length and moves downstream with increasing turbulence intensity. Furthermore, a corresponding reduction in the chordwise extension of the separation bubble is seen before it disappears at $TI = 5.6$ %. The time-averaged results showed a decrease in the drag coefficient with increasing $TI$. A more detailed analysis revealed that the contribution of the pressure component decreased due to the reduction in the length of the separation bubble, while that of the friction component increased with increasing inflow $TI$. In the current study on the flow around the thicker (20 % thickness) LM45 airfoil, the separation bubble moves slightly downstream with increasing $TI$ before disappearing at $TI = 11.2$ %. However, here the length of the separation bubble does not decrease with increasing $TI$. The absence of a separation bubble at $TI = 11.2$ % is due to the increased momentum exchange within the boundary layer with the flow being transitional and closer to the turbulent regime than the laminar regime (discussed in Section 4.3) at the location, where it would have otherwise separated. Correspondingly, a resulting increase of the drag coefficient with increasing $TI$ is seen.

In the study by Breuer (2018) a decrease in the lift coefficient is observed with an increase in $TI$ up to 5.6 % before it stays constant. It is known that a separation bubble close to the leading edge could increase the lift coefficient due to the increase in the apparent camber caused by the presence of the separation bubble. With increasing $TI$ and the downstream shift of the separation bubble, the lift coefficient then decreases. In the present study, the lift coefficient increases with increasing $TI$, however, very slightly (a relative change of 3 %) and is likely caused by the slight downstream shift of the separation region, which increases the extent of the laminar flow along the chord.

A combination of these factors results in an increasing lift-to-drag ratio with increasing $TI$ in Breuer (2018), whereas the lift-to-drag ratio in the current study reduces.

## 4.2 Influence of inflow turbulence intensity on transition

To investigate the influence of the inflow turbulence intensity on the onset of transition $x_{tr}/c$, several methods exist in the literature. In case of attached flows, the transition from laminar to turbulent flow is easily distinguishable by a sudden and strong increase in the boundary-layer thickness or from the shear stress near the wall. The normalized Reynolds shear stress $\overline{u'v'}/u_\infty^2$ can be considered for the determination of the onset of transition as it is a quantity defining the exchange of momentum into the boundary layer. For flows with a laminar separation bubble, Yuan et al. (2005) defined the onset of transition as the point where the normalized Reynolds shear stress $-\overline{u'v'}/u_\infty^2$ reaches a value of 0.001 and demonstrated a clearly visible rise. Figures 6a and 6b show contour plots of the Reynolds shear stress at an inflow $TI$ of 0 % and 11.2 %, respectively. An obvious upstream shift in the onset of transition can be observed for rising $TI$. A plot showing the Reynolds shear stress at the height of the

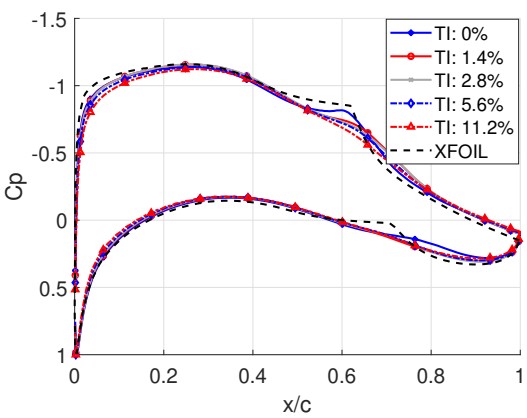

(a) Pressure coefficient based on the mean flow.

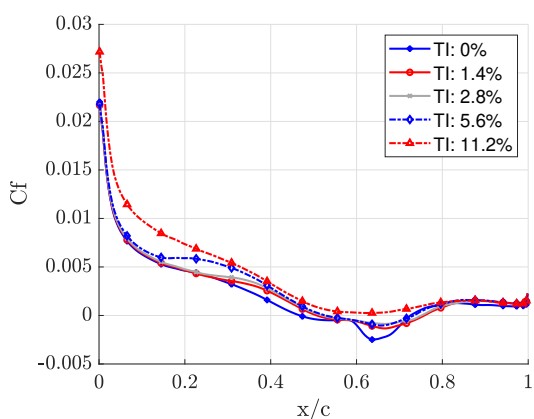

(b) Suction side friction coefficient based on the mean flow.

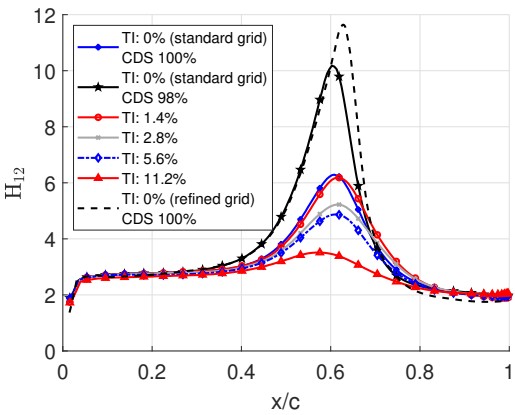

(c) Shape factor $H_{12}$ along the suction side.

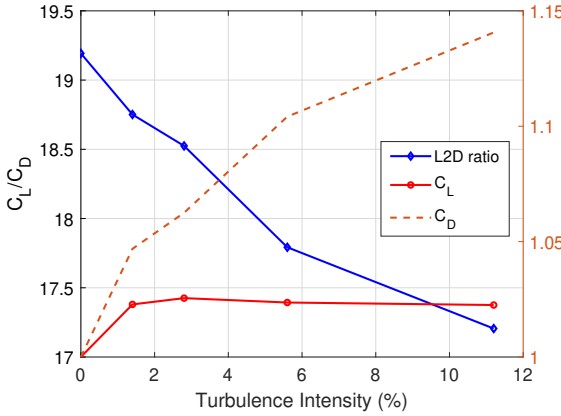

(d) Lift-to-drag ratio. Right axis scaled by ref. case with $TI = 0$ %.

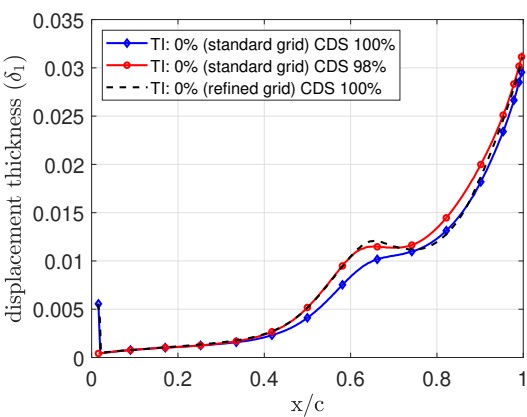

(e) Displacement thickness $\delta_1$ along the suction side.

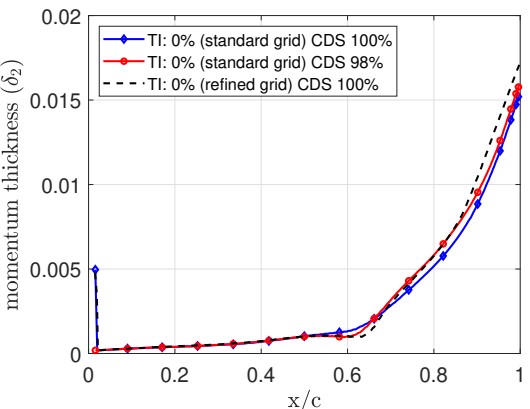

(f) Momentum thickness $\delta_2$ along the suction side.

**Figure 5.** Distribution of aerodynamic properties of the airfoil at a Reynolds number of $10^5$ and $\alpha = 4°$.

**Table 3.** Location of the separation and reattachment points of the averaged flow for different inflow turbulence intensities extracted at the first wall-normal cell; transition onset based on two different criteria.

| Turbulence Intensity $TI$ | Transition Onset $x_{tr}/c$ | | Separation point $x_{sep}/c$ | Reattachment point $x_{rea}/c$ |
|---|---|---|---|---|
| | ($-\overline{u'v'}/u_\infty^2 = 0.001$) (Yuan et al., 2005) | max $H_{12}$ (Burgmann and Schröder, 2008) | | |
| 0 (refined grid, 100 % CDS) | 0.508 | 0.626 | 0.489 | 0.717 |
| 0 (standard grid, 100 % CDS) | 0.495 | 0.608 | 0.506 | 0.738 |
| 0 (standard grid, 98 % CDS) | 0.518 | 0.602 | 0.486 | 0.727 |
| 1.4 % | 0.435 | 0.618 | 0.512 | 0.759 |
| 2.8 % | 0.390 | 0.618 | 0.523 | 0.754 |
| 5.6 % | 0.355 | 0.613 | 0.525 | 0.735 |
| 11.2 % | 0.320 | 0.570 | - | - |

displacement thickness at different inflow $TI$ is depicted in Figs. 6c and 6d. Obviously the transition onset moves upstream with rising $TI$. A brown dashed line marks the location, where the threshold $-\overline{u'v'}/u_\infty^2 = 0.001$ is reached. Furthermore, Table 3 summarizes the location of the transition onset based on this criterion for the different cases simulated. However, the definition of a threshold is arbitrary and this method yields the position of transition onset upstream of the separation point. Nonetheless, the results of the threshold method are qualitatively meaningful as it shows a clear upstream shift in the onset of

transition even if the threshold value of $-\overline{u'v'}/u_\infty^2$ would be increased as seen in Fig. 6d.

Another method for the determination of transition onset is based on the investigation of the shape factor $H_{12}$. The position of transition onset is defined as the location of the maximum value of $H_{12}$ along the surface as already discussed in Section 4.1. The location of this transition onset is also included in Table 3. Using this method, the location of transition onset is within the separation bubble in the cases with laminar separation and is therefore a more reasonable approach for the current study.

For a better visualization of the transition process Fig. 7 displays instantaneous snapshots of the dimensionless $Q$-criterion normalized by $u_\infty^2/c^2$ and colored by the mean streamwise velocity averaged along the spanwise direction and in time. At an inflow turbulence intensity of 0 %, spanwise rolls which are characteristic for an inflectional instability with spanwise vortices are clearly visible. At their onset, they are more or less two-dimensional. As the flow progresses downstream, slightly three-dimensional effects are seen. This is in agreement with a study by Rao et al. (2014) where the inviscid instability interacts with

viscous instabilities closer to the wall leading to fully turbulent flow.

Similar spanwise rolls are also visible at an inflow turbulence intensity of 1.4 %, albeit occurring with relatively more three-dimensional effects. One reason for the increased three-dimensional effects is the presence of Klebanoff streaks within the boundary layer. These streaks are further discussed and illustrated in Section. 4.4. With an increase of the turbulence intensity to 2.8 and 5.6 % the influence of boundary layer streaks becomes even more pronounced as will be discussed in Section 4.3.

However, the presence of spanwise rolls is still apparent.

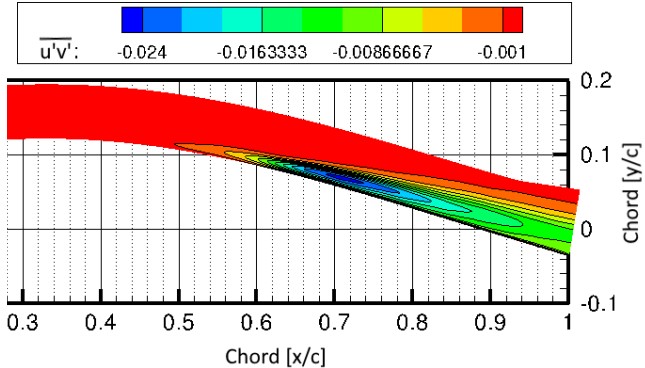

(a) Contour of the averaged Reynolds shear stress $-\overline{u'v'}/u_\infty^2$ at an inflow $TI$ of 0 %. Transition onset is seen at 0.495 chord.

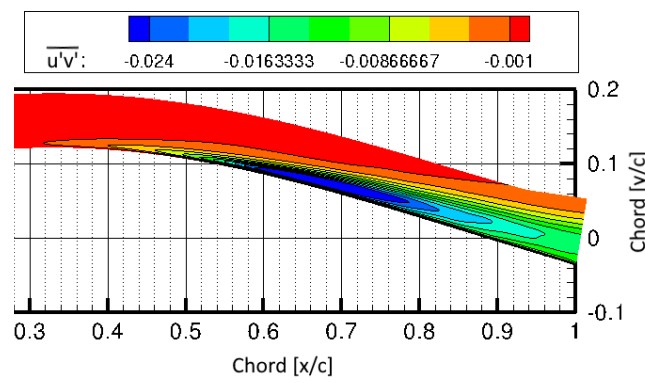

(b) Contour of the averaged Reynolds shear stress $-\overline{u'v'}/u_\infty^2$ at an inflow $TI$ of 11.2 %. Transition onset is seen at 0.320 chord.

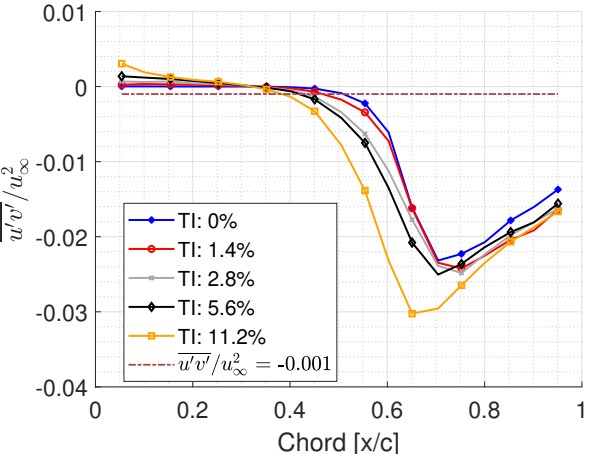

(c) Normalized averaged Reynolds shear stress for varying $TI$ at the height of the displacement thickness.

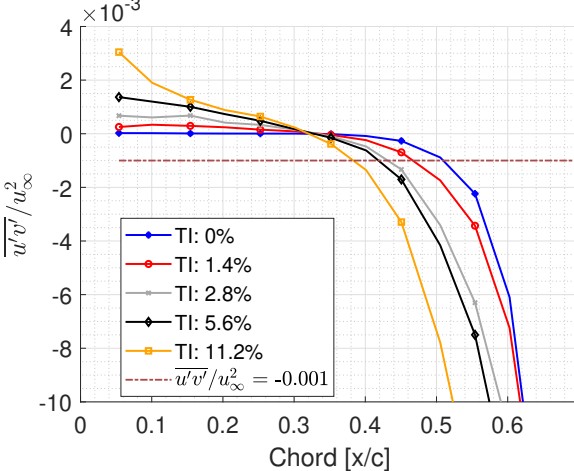

(d) Zoom into Fig. 6c showing the upstream shift in the onset of transition with rising $TI$.

**Figure 6.** Contours and profiles of the averaged Reynolds shear stress showing the upstream shift in the onset of transition according to the transition criterion based on the threshold of Reynolds shear stress ($-\overline{u'v'}/u_\infty^2 = 0.001$) (Yuan et al., 2005).

For the case with a very high turbulence intensity of 11.2 %, the flow does not separate and spanwise rolls are no longer present while the streaks take over the transition process.

Similar observations with a three-dimensionality of the separated flow and spanwise waviness of the separated shear layer with increasing inflow turbulence has been made in studies by Lardeau et al. (2012), Rao et al. (2014), Scillitoe et al. (2019) and Zaki et al. (2010). Zaki et al. (2010) showed that at relatively high inflow turbulence intensity ($> 6.5$ % for their studied


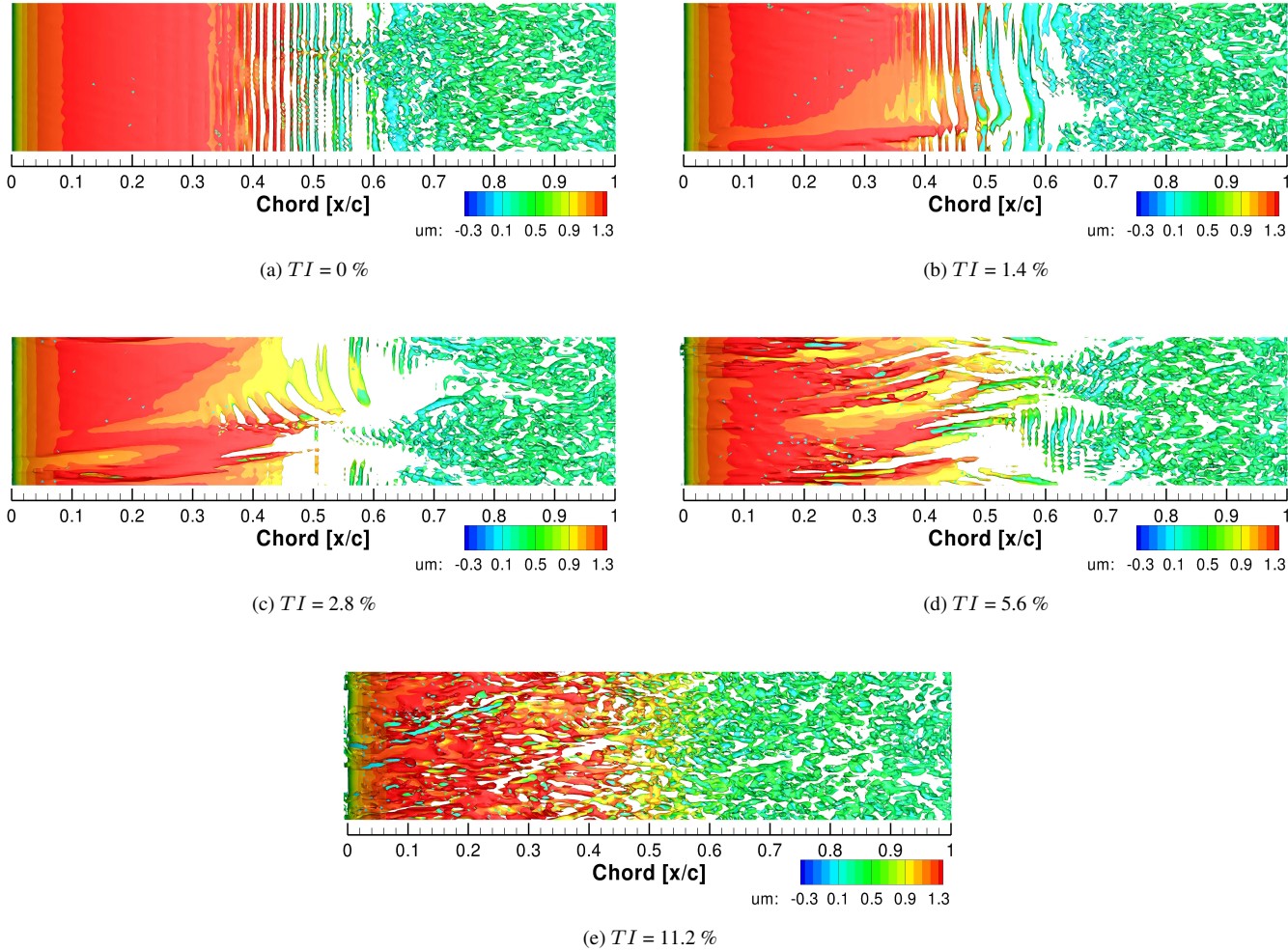

(a) $TI = 0\ \%$

(b) $TI = 1.4\ \%$

(c) $TI = 2.8\ \%$

(d) $TI = 5.6\ \%$

(e) $TI = 11.2\ \%$

**Figure 7.** Instantaneous iso-surfaces of the $Q$-criterion ($Q = 250$) colored by the mean streamwise velocity normalized by $u_\infty^2/c^2$.

case) bypass transition precedes separation and maintains an attached turbulent flow, which is similar to the case with an inflow $TI$ of 11.2 % presented here where the transitional region prevents flow separation.

Figure 8 shows the development of the RMS values of the streamwise velocity fluctuations $u' = (u(x,y,z,t) - \overline{u}(x,y))/u_\infty$, i.e., averaged along the spanwise direction and in time at a location corresponding to the boundary layer displacement thickness. In the case without inflow turbulence the growth of the fluctuations follows a somewhat exponential trajectory (linear on the log-scale) beginning at about 30 % chord. This corresponds to the location that marks the beginning of an adverse pressure gradient (APG) as seen in Fig. 5a. An APG could promote the amplification of inflectional instabilities resulting in large growth rates of the $u'$ disturbance as seen in Fig. 8a. The presence of an inflectional instability within the boundary layer will be shortly discussed below. The linear growth seen in Fig. 8a continues until the flow turns fully turbulent as will be shown in Section 4.3.

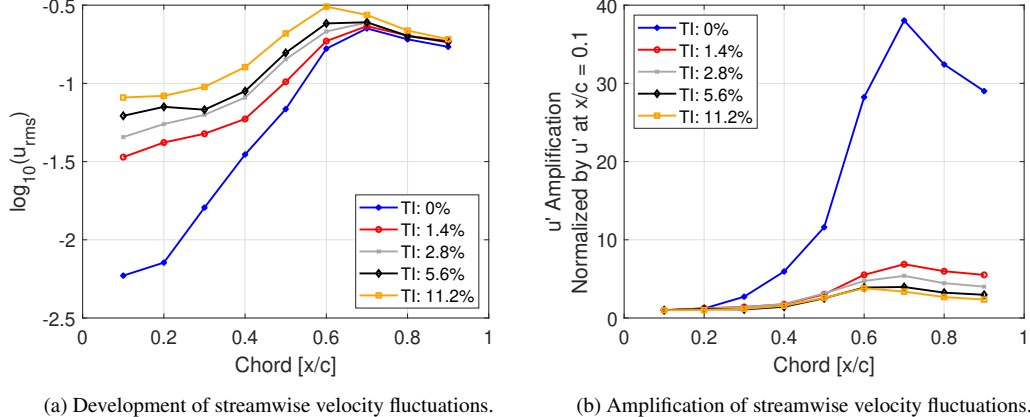

(a) Development of streamwise velocity fluctuations.

(b) Amplification of streamwise velocity fluctuations.

**Figure 8.** Development and amplification of RMS values of the streamwise velocity fluctuations at different inflow turbulence intensities determined at the height of the displacement thickness.

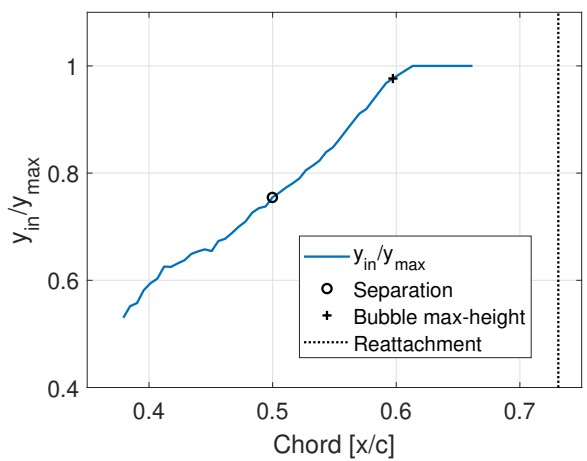

**Figure 9.** Ratio of the location of the inflection point to the location of the peak production of turbulent kinetic energy of the disturbances indicating the increasing importance of the inflectional mode.

At inflow turbulence intensities of 1.4 % to 5.6 % the RMS values of $u'$ are obviously elevated in comparison with the case without inflow turbulence on account of the receptivity of the boundary layer to external disturbances. The influence of the APG on the growth of inflectional instabilities is observed, similar to the case without inflow turbulence. The APG in addition to the growth of inflectional instabilities also allows for the growth of Klebanoff modes (Zaki et al., 2010) which, if present, could lead to turbulent bursts as seen in Section 4.4. With a rise in inflow turbulence intensity to 11.2 % the growth in $u'$ is no

longer associated with an inflectional instability, but is attributed to the growth of Klebanoff modes.

Figure 8b depicts the relative growth or amplification of $u'$ normalized by $u'$ at the location $x/c = 0.1$ for each case. The data are extracted at the height of the displacement thickness. It must be noted that an obvious decrease in the relative amplification factor with increasing $TI$ is seen with a clear differentiation between the cases with added inflow turbulence ($TI = 1.4\,\%$ and higher) compared to the case without inflow turbulence. The reduced relative increase in the amplification factor in cases with added inflow turbulence is due to the receptivity of the boundary layer to disturbances and the presence of Klebanoff streaks that have developed within the boundary layer which are already present at 10 % chord, i.e., the point used for the normalization. It is interesting to note that there is a drop in the amplitude of $u'$ from 60 to 70 % chord in case of an inflow turbulence intensity of 5.6 and 11.2 %, while the drop is seen between 70 and 80 % chord in the other cases. If this is used as an indication for transition to turbulent flow, it is once more clear that transition does move upstream with increasing turbulence intensity.

In the absence of added inflow turbulence, an inflectional profile of the averaged streamwise velocity is observed from approximately 38 % up to 66 % chord followed by the reattachment of the fully turbulent flow at 73.8 % chord. It is interesting to note that an inflectional instability is seen upstream of the separation point (50.06 % chord) and is convected downstream into the separation region, where a K-H type of instability develops near the maximum separation bubble height (59.73 % chord) and finally breaks down to turbulent flow. This observation is in good agreement with the experimental and theoretical study by Diwan and Ramesh (2009) where an exponential growth rate of disturbances according to an inflectional instability upstream of the separation region is followed by a K-H type of instability in the separation bubble. Using a method similar to that used by Diwan and Ramesh (2009) to determine the importance of the contribution of the inflectional mode to the disturbance dynamics, a ratio of $y_{in}$ to $y_{max}$ is plotted in Fig. 9, where $y_{in}$ is the wall-normal distance at which the inflectional point of the mean streamwise velocity is located and $y_{max} = \max(\overline{u'v'}\,\frac{\partial U}{\partial y})$ is the location of the peak production of the turbulent kinetic energy of the disturbance. As discussed in Diwan and Ramesh (2009) if the ratio tends towards zero, the wall mode is stronger and as the ratio approaches unity the inflectional mode is dominant. For a separated profile this value is expected to be unity while it is expected to be zero for a Blasius profile since $y_{in} = 0$. This behavior is visible in Fig. 9 with the ratio approaching unity just prior to the maximum height of the separation bubble. This is again similar to the experimental data of Diwan and Ramesh (2009), where the ratio approached unity after separation. Furthermore, the onset of the inflectional instability at 38 % chord is in good agreement with the observations of spanwise rolls as seen in Fig. 7a indicating that these rolls are inflectional in nature even though they are presently upstream of the separation.

### 4.3 Analyzing transition based on power spectral densities

To further investigate the mode and process of transition as well as the influence of the turbulence energy spectra on the boundary layer receptivity in case of bypass transition through the formation of boundary layer streaks, it was necessary to analyze the power spectral densities along the chord for the different inflow turbulence intensities under consideration. Figure 10 shows the turbulent kinetic energy PSD plots computed using a Hann windowing function on the data which were collected at every 10 % of the chord. A total of $2^{16}$ data points covering 3.28 dimensionless time units was used. In order

to follow the transition process even in the laminar separation region, data evaluations were carried out at the height of the boundary layer displacement thickness and at the mid-span. Figure 10f indicates the locations where these data are evaluated.

At an inflow turbulence intensity of 0 % the growth of inflectional instabilities (see disturbance velocity in Fig. 9) is apparent at about 30 % chord as discussed in Section 4.2. It must be noted that spanwise rolls were already seen at 30 % chord at some instances in time, thus making their presence visible in the PSD plot. This corresponds to the green solid line in Fig. 10a. A clear increase in energy up to thousand dimensionless units is recognized. The inflectional instability continues to grow in the downstream direction. This growth is seen at 40 % chord without any noticeable irregularity in the PSD plot compared to the upstream location. However, at 50 % chord, a further increase in energy between approximately 70 and 150 dimensionless units is found, indicated by the purple solid line. This increase in energy corresponds to the location (at some instants in time) of breakdown of the roll-up, through the Kelvin-Helmholtz mechanism. This is in a similar dimensionless frequency range seen in the experiment by Boutilier and Yarusevych (2012) for the separated shear layer on the NACA 0018 profile at Re = 100k and a similar angle of attack.

At an inflow turbulence intensity of 1.4 %, the growth of inflectional instabilities beginning at about 30 % chord is visible in Fig. 10b. Here, again, the disturbances continue to grow in the downstream direction at 40 and 50 % chord recognized by the increase in energy up to thousand dimensionless units. At 70 % chord the flow is fully turbulent indicated by the rise in energy in the PSD plot, similar to that at the downstream locations. It is obvious that transition to turbulence takes place further downstream than in the case with $TI = 0$ % where the flow is turbulent close to 60 % chord, as seen in the PSD plot. This unexpected downstream shift of the transition onset with an increase of $TI$ from 0 to 1.4 % is caused by the presence of a high-speed boundary layer streak which delays separation (see discussion in Section 4.4) near the mid-span for considerably large parts of the time period during which the data were collected.

At an increased inflow turbulence intensity of 2.8 %, a rise in energy up to thousand dimensionless units is already visible at 20 % chord in the PSD plot in Fig. 10c which is similar to that observed at lower $TI$ in the presence of inflectional instabilities indicating an earlier growth of these instabilities beginning within the region of the FPG. In this case, spanwise rolls were not visible by a Q-criterion analysis, but waves which eventually lead to the formation of spanwise rolls were noticed at some instances. Similar waves are seen, at 30 % chord in the case without inflow turbulence. At 40 % chord a further increase in energy is observed, similar to that seen at 50 % chord in Fig. 10a. This corresponds to the breakdown of the K-H rolls.

At an inflow turbulence intensity of 5.6 %, a noticeable increase in energy up to around 100 dimensionless units is seen in Fig. 10d when compared to the cases with lower inflow turbulence intensities. This is attributed to the presence of streaks within the boundary layer which will be discussed in Section 4.4. At an inflow $TI$ of 2.8 % a similar effect, albeit with somewhat reduced energy is seen at 10 % chord. The increased energy at $TI = 5.6$ % is due to the increased frequency of streak formation within the boundary layer observed for increasing inflow turbulence, thus making their presence more prominent with increasing $TI$. It must be noted that the data for the generation of the PSD was collected over a period of 3.28 dimensionless units, therefore, it is not unexpected to see both, the influence of streaks and inflectional instabilities simultaneously in the PSD plots. For instance, at a $TI$ of 2.8 %, inflectional instabilities dominate while at $TI = 5.6$ % an increase in energy at 20 % chord up to thousand dimensionless units indicates the presence of an inflectional type of instability. Note that this increase in

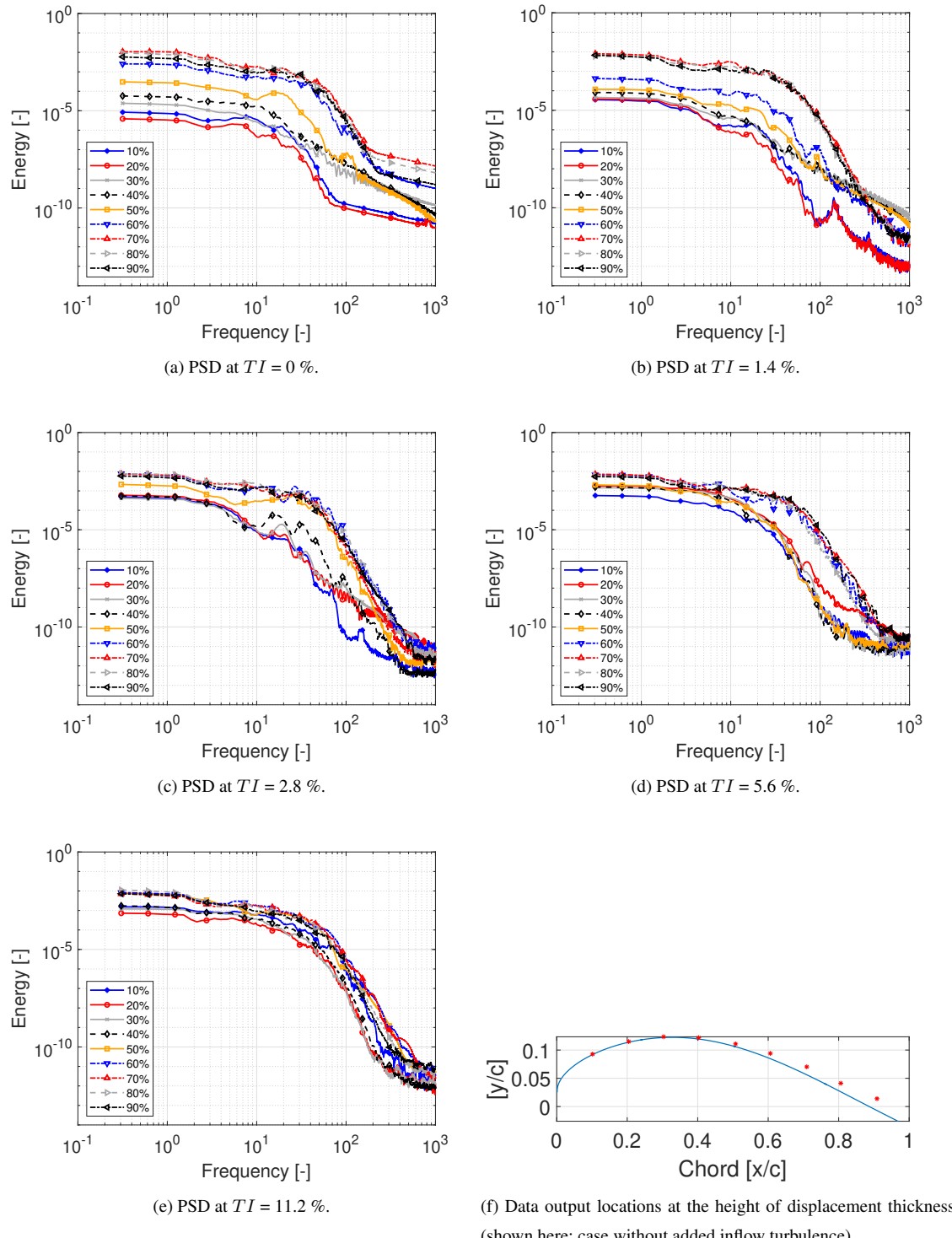

(a) PSD at $TI$ = 0 %.

(b) PSD at $TI$ = 1.4 %.

(c) PSD at $TI$ = 2.8 %.

(d) PSD at $TI$ = 5.6 %.

(e) PSD at $TI$ = 11.2 %.

(f) Data output locations at the height of displacement thickness (shown here: case without added inflow turbulence).

**Figure 10.** Power spectral density at different $TI$ determined at the height of the displacement thickness in the mid-span at every 10 % chord.

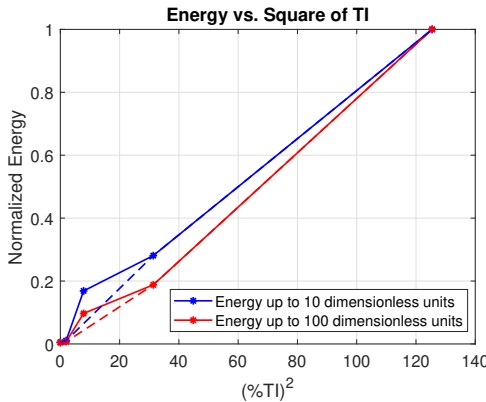

**Figure 11.** Energy at 10 % chord up to 10 and 100 dimensionless frequency units against the square of the turbulence intensity. The energy is normalized by the value at a $TI$ of 11.2 %.

energy is not present downstream of this chord position due to the increased contribution of streaks to the transition process. An increase in energy up to approximately 100 dimensionless units up to 50 % chord is seen before the flow turns fully turbulent between 60 and 70 % chord indicating the dominance of boundary layer streaks.

At an inflow $TI$ of 11.2 % the energy content in the vicinity of the leading edge, i.e., as seen at 10 % chord and further downstream in Fig. 10e, is similar to the energy level in the transition region at $TI = 0$ %. At this highest $TI$ level of 11.2 % no clear laminar region can be distinguished in the PSD plot. The energy levels are comparable to those of the transitional region of lower $TI$ inflow cases. The increase in energy from transition to turbulence, however, is clearly visible at 60 % chord and further downstream.

One of the key objectives of running a PSD analysis was to study the receptivity of the boundary layer in response to increasing $TI$. An experiment by Fransson et al. (2005) showed that the initial energy in the boundary layer is proportional to the square of the turbulence intensity. Their study included FST between 1.4 and 6.7 % over a flat-plate. A similar analysis was carried out here, with the total energy plotted against the square of $TI$ as seen in Fig. 11. Since the boundary layer is receptive to low frequencies, especially near the leading edge, it was decided to plot the total energy including the frequency content up to 10 dimensionless frequency units and at 10 % chord. Since the influence of streaks was observed up to 100 dimensionless frequency units (see Section 4.4), a second line with the energy including the content up to 100 dimensionless units is plotted. In both cases, ignoring the anomaly at an inflow $TI$ of 2.8 % (the dashed lines ignore this data point), a proportionality via a nearly linear relationship between the energy content and the square of $TI$ is seen. This is in good agreement with the study by Fransson et al. (2005).

### 4.4 Streak analysis

As discussed in Section 4.3 a noticeable increase in energy up to around 100 dimensionless units is seen at an inflow $TI$ of 5.6 % compared to the cases with lower inflow $TI$. It is assumed that this is due to the relative increase in the formation of

streaks within the boundary layer such that they can be detected in the PSD plot. This is now investigated based on the case of $TI = 5.6$ %.

Due to the formation of multiple streaks within the boundary layer during the simulation, it was found necessary to isolate, extract and analyze data prior to the entry of a selected streak and compare the generated PSD plot to a PSD plot generated during the passing of the streak through the region of interest. This is done in order to study the frequency characteristics of the streaks. For this purpose, data were extracted over $300,000$ time steps covering an interval of 1.5 dimensionless time units. The data are sampled at every tenth time step, resulting in $30,000$ data points. This is a sufficiently large number of data points allowing for the resolution of low frequencies. According to this data, the streak as seen in Fig. 12a passes through 10 % chord at around the $12,000^{th}$ data point. A comparison between the PSD plots at this chord location before and after the passing of the streak is depicted in Fig. 12b. An increase in energy up to 100 dimensionless units is seen. Similar increases in energy up to 100 dimensionless units are also detected when the streak passed through downstream chord positions.

Thus, the increase in energy up to 100 dimensionless units in the PSD plots of Section 4.3 is indeed an indication for the presence of streaks. It must be noted that streaks are clearly visible within the boundary layer also at lower inflow $TI$ and near the leading edge as seen in Fig. 13, which shows boundary layer streaks at inflow turbulence intensities of 1.4 and 11.2 % at a wall-normal location corresponding to the displacement thickness at 20 % chord.

As stated earlier and to clarify this issue, the rate of formation of streaks is much lower at low $TI$ which is obvious by comparison of Fig. 13a with Fig. 13b. Therefore, an obvious indication of their presence is not seen in the PSD plots at lower $TI$.

Just as in the study by Hack and Zaki (2014) streaks of different sizes, amplitudes and orientations were found in the collected data and at different inflow $TI$. Additionally, the maximum spanwise dimension of the streaks decreased with increasing $TI$. This is in line with the findings of Zaki et al. (2010).

At low inflow turbulence intensities, for example at 1.4 and 2.8 % the time-averaged results summarized in Table 3 indicate that the free-stream turbulence has negligible effects on the onset of separation. However,the instantaneous perturbation fields shown in Fig. 14 show that boundary layer streaks affect the instantaneous separation point. The black rectangular box marks the location of the mean separation region. Instantaneous separation as indicated by the slightly translucent orange iso-surface is shifted upstream in the presence of negative perturbation velocity (dark streaks) and downstream in the presence of positive perturbation velocity (light streaks). These results are in line with earlier studies (Scillitoe et al., 2019; Zaki et al., 2010). Tangermann and Klein (2020) also observed the influence of inflow turbulence on the separation bubble. The separation region was found to be delayed in some spanwise regions compared to the case without added turbulence.

On account of an APG, the inner mode of instability (Vaughan and Zaki, 2011) was observed at all inflow turbulence intensities. A common scenario attributed to this mode is a local overlap of the trailing edge of a low-speed streak and the leading edge of a high-speed streak which results in the wall-normal velocity profile to become inflectional. This leads to the development of a predominantly varicose instability near the wall. This scenario is the one observed in the present simulations. That is illustrated in Fig. 15 for the case with an inflow $TI$ of 2.8 %. Each image is taken at a dimensionless time period of 0.1 units apart and at a wall-normal distance corresponding to the displacement thickness at 20 % chord. Fig. 15a marks the

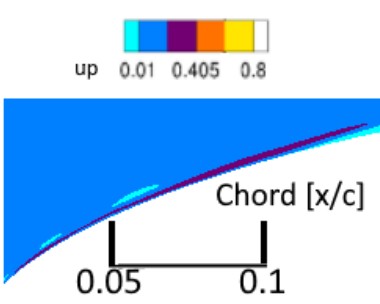

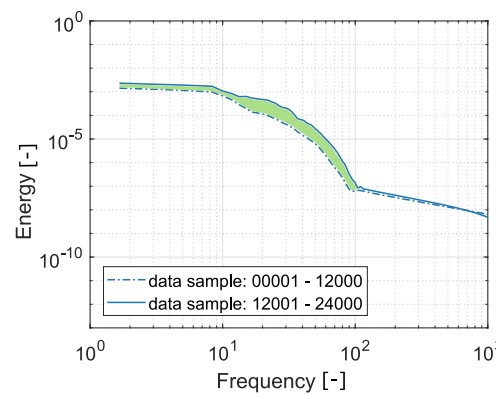

(a) A streak passing through 10 % chord.

(b) 10 % chord. The dashed line is the PSD prior to the passing of a streak and the solid line is the PSD on the passing of the streak. The shaded region marks the energy difference.

**Figure 12.** Influence of a boundary layer $u'$ streak on the PSD plot at $TI = 5.6$ %.

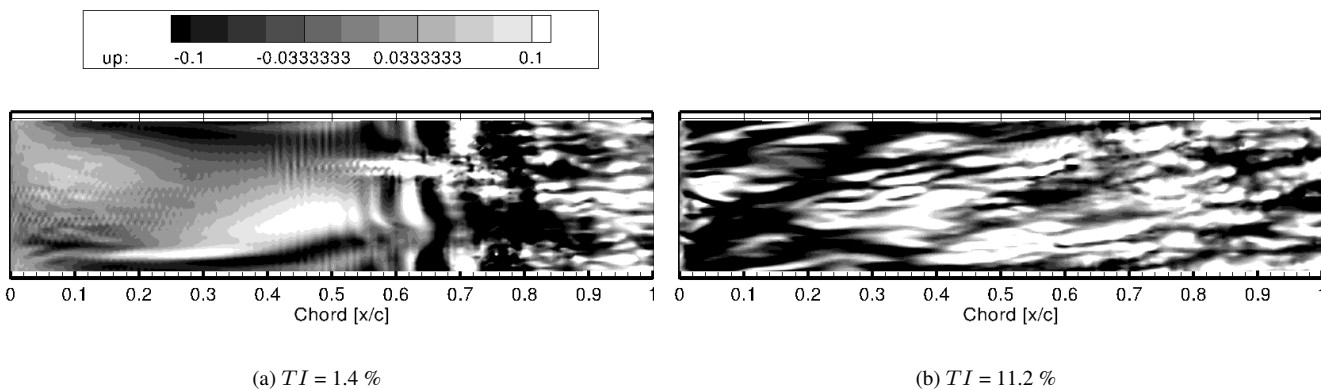

(a) $TI = 1.4$ %

(b) $TI = 11.2$ %

**Figure 13.** $u'$ boundary layer streaks. Slices are taken at a wall-normal height equal to the displacement thickness at 20 % chord.

early stages of the overlap between the light high-speed streak and the dark low-speed streak. The overlap takes place with the lower of the two marked streaks. The two streaks have been marked to distinguish them as two separate streaks, a fact that was confirmed by going back in time to the formation of the individual streaks. In Fig. 15b the varicose inner instability is seen. At the onset of its formation, the instability has the same spanwise dimension as the parent streak. A similar observation was made in Scillitoe et al. (2019). As the instability develops, the varicose formation becomes clearer and is highlighted in Fig. 15c. After further 0.1 dimensionless time units, the instability breaks down to form a turbulent spot as seen in Fig. 14b. This turbulent spot grows and progresses downstream to finally join the fully turbulent flow.

As mentioned above, the origin of the inner instability mode is an overlap of the leading edge of a high-speed streak with the trailing edge of a low-speed streak. An investigation into the formation of the instability must therefore take both streaks into

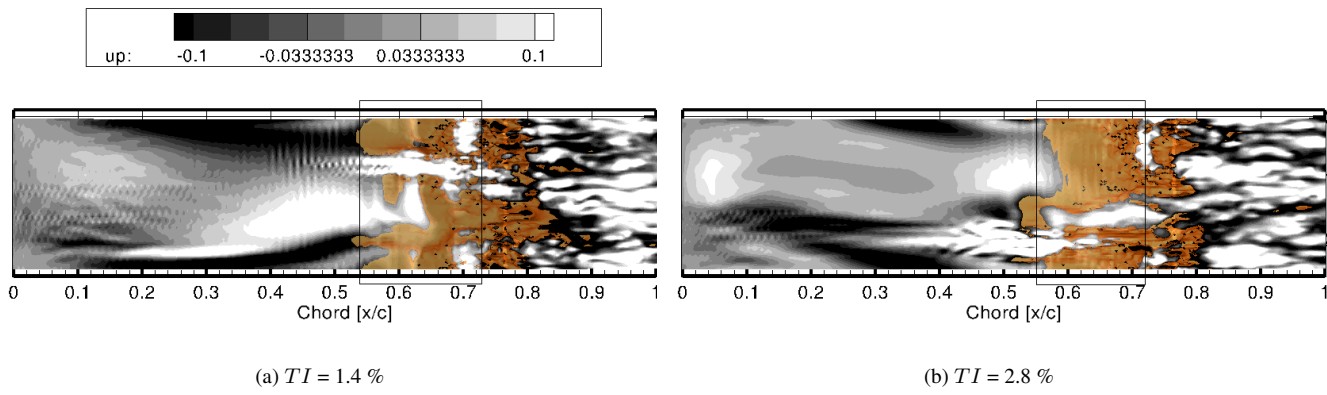

(a) $TI = 1.4\,\%$

(b) $TI = 2.8\,\%$

**Figure 14.** Influence of boundary layer streaks on instantaneous separation. The light and dark streaks represent high- and low-speed streaks, respectively. The orange iso-surface represents instantaneous separation. Slices are taken at a wall-normal distance corresponding to the displacement thickness at 20 % chord.

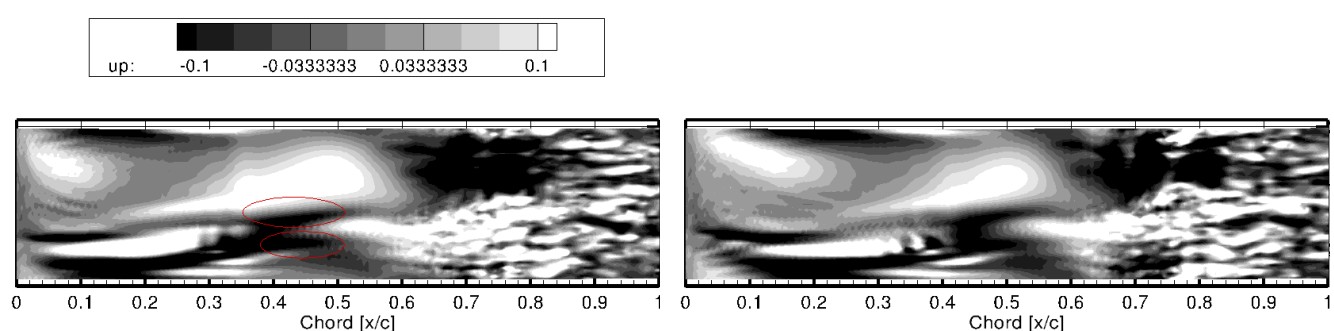

(a) Early stages of overlap between the high- and low-speed streak. Two different low-speed streaks are marked. The lower of the two is the one of interest.

(b) Formation of the varicose instability.

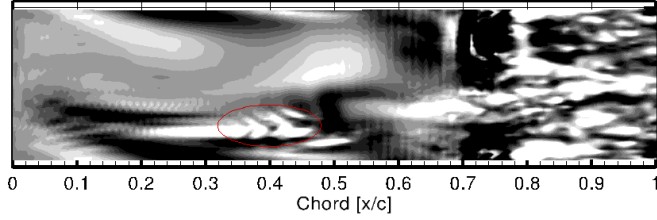

(c) Growth of the instability.

**Figure 15.** Development of an inner type of instability mode at $TI = 2.8\,\%$. Contours show the tangential velocity perturbations $u'$. Each image in this sequence is taken at a dimensionless time period of 0.1 units apart and at a wall-normal distance corresponding to the displacement thickness at 20 % chord.

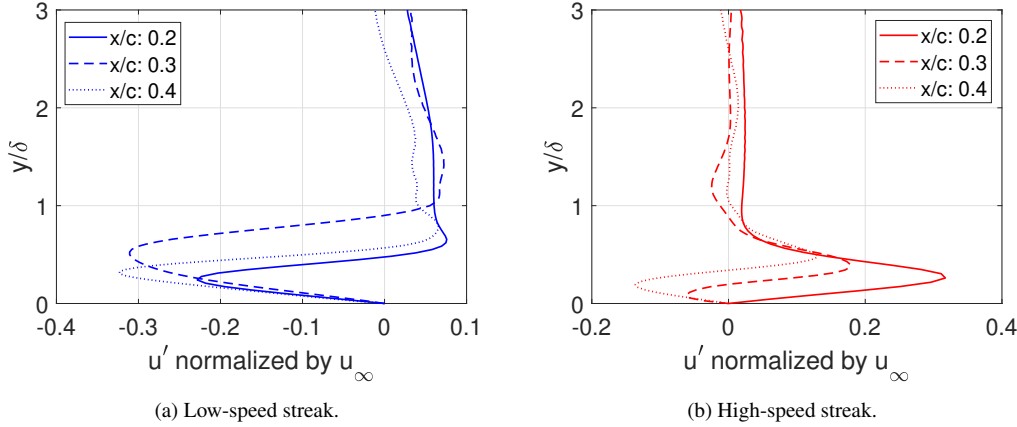

(a) Low-speed streak.          (b) High-speed streak.

**Figure 16.** $u'$ streaks that lead to the formation of an inner instability at $TI = 2.8$ %.

account. Fig. 16 shows the corresponding $u'$ perturbations of the low-speed streak (Fig. 16a) which is in time overlapped by
the high-speed streak (Fig. 16b). For this purpose, consider a slowly moving low-speed streak upstream of which a high-speed
streak has formed. The high-speed streak catches up and overlaps the trailing edge of the low-speed streak. This phenomenon
causes an inflectional instability. The profiles were generated over a sequence of time steps. The perturbation profile chosen to
be plotted is that which possesses the maximum perturbation velocity on the passing of the two relevant streaks.

As seen in Fig. 16a, the negative perturbation velocity increases as the streak progresses downstream and the streak rises
within the boundary layer from 20 to 30 % chord, which is to be expected for a low-speed streak. Even though a slight increase
in the negative perturbation velocity is seen at 40 % chord, the streak oddly drops within the boundary layer while it is expected
to rise. The time instant of this perturbation velocity plot corresponds to an instant where the varicose instability has already
begun to develop upstream of the streak due to shear from the overlap of the high-speed streak. This effectively alters the streak
and is a possible cause for this behavior. Fig. 16b shows the perturbation velocity due to the high-speed streak. In this case, the
streak has a typical form of a high-speed streak at 20 % chord, but begins to overlap with the low-speed streak at about 30 %
chord which is evident from the points of inflection. The streak amplitude increases up to 20 % chord reaching a value of 32 %
of the free-stream velocity prior to the development of the instability. Similar values leading to the development of the inner
instability are found at other locations and other $TI$ values. This is in agreement with the work by Andersson et al. (2001),
who observed the critical streak amplitude to be 37 % of the free-stream velocity for the varicose mode.

It must be noted that no evidence was found for an outer type of instability mode. This is attributed to the relatively strong
APG on the suction side (Marquillie et al., 2011), while it is known that a zero- or favorable pressure gradient is decisive for
such a mode (Jacobs and Durbin, 2001).

## 5 Conclusions

To further optimize the design of wind turbine blades, the knowledge on the transition process on the blades has to be thoroughly
improved. For typical turbulence intensities of atmospheric turbulence it is expected that bypass transition plays an important role. Based on wall-resolved LES predictions the present study should help to expand the knowledge base and to pave the way for the simulation of realistic Reynolds numbers. Furthermore, this study includes an elaborate spectral analysis of the transition process, a method not often employed by other similar studies. The LES predictions for the flow around the wind turbine blade of the type LM45.3p at $\mathrm{Re}_c = 10^5$ and $\alpha = 4°$ for varying inflow turbulence intensities yield the following results:

- A separation bubble roughly between 50 and 75 % of the chord is observed for turbulence intensities $TI \leq 5.6$ %. At $TI = 11.2$ % the separation vanishes. This observation is in partial agreement with previous investigations (Breuer, 2018) on much thinner airfoils (SD7003: 8.51 % relative thickness), where the separation bubble already completely disappears at $TI \geq 5.6$ % at a comparable $\mathrm{Re}_c$ of 60,000. This is also in good agreement with another study (Zaki et al., 2010), where separation was seen up to $TI = 6.5$ % at a Reynolds number of 138,500. The frequency range of
the detected separated shear layer between 70 and 150 dimensionless units is similar to the experiment by Boutilier and Yarusevych (2012) on the NACA 0018 airfoil at a similar angle of attack and Reynolds number.

- The analysis of the amplification of the velocity fluctuations $u'$ at $TI = 0$ % shows an exponential growth, which begins at the onset of the APG. It is found that the primary instability mechanism within the separation bubble is inflectional in nature and its origin can be traced back to the region upstream of the separation. The instability is also clearly visible
in the power spectral density of the turbulent kinetic energy. These findings are in agreement with the experimental and theoretical study of Diwan and Ramesh (2009).

- At inflow turbulence intensities up to $TI = 5.6$ % the presence of the inflectional instability is noticed in the turbulent kinetic energy power spectral density plots. At $TI = 11.2$ % there is no such indication and the transition process is completely governed by boundary layer streaks. With rising $TI$ the influence of boundary layer streaks increases.
The transition mode changes from an inflectional instability dominated one due to the APG to a transition mechanism influenced by the presence of streaks within the boundary layer.

- It is found that the boundary layer is receptive to external disturbances such that the initial energy within the boundary layer is proportional to the square of the turbulence intensity. A similar finding was made in an experimental study by Fransson et al. (2005) over a flat plate.

- In the presence of negative velocity perturbations describing low-speed streaks, separation is found to be shifted upstream. Contrarily, in the presence of positive velocity perturbations according to high-speed streaks separation is shifted downstream. These results are in line with earlier studies (Scillitoe et al., 2019; Zaki et al., 2010; Tangermann and Klein, 2020). Consequently, it is obvious that boundary layer streaks affect the instantaneous separation point.

- In the presence of boundary layer streaks, an inner type of instability mode is observed whereby the leading edge of a high-speed streak overlaps the trailing edge of a low-speed streak resulting in an inflectional profile and a varicose mode of instability. This is expected due the APG (Marquillie et al., 2011). The critical streak amplitude was found to be about 32 % of the free-stream velocity. This is in good agreement with the observations by Andersson et al. (2001), who reported the critical streak amplitude to be 37 % of the free-stream amplitude.

Overall, the results show that the applied methodology of wall-resolved LES with injected inflow turbulence works reliably and provides physically meaningful results. In follow-up studies the Reynolds number will be successively increased in order to understand the transition scenario at the real case under varying inflow conditions.

*Author contributions.* BAL performed all simulations, post-processed and analyzed the simulation data and wrote the draft version of the paper. APS and MB supervised the investigations, gave technical advice in regular discussions and improved the paper and its revision.

*Competing interests.* The authors declare that they do not have any conflicts of interest.

*Acknowledgements.* The project is financially supported through the EKSH-Promotionsstipendien program. The authors acknowledge the North-German Supercomputing Alliance (HLRN) for providing HPC resources that have contributed to the research results reported in this paper. We acknowledge financial support by Land Schleswig-Holstein within the funding programme Open Access Publikationsfonds.

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
