# Peer review of "Investigation Into Boundary Layer Transition Using Wall-Resolved LES and Modeled Inflow Turbulence"

_Wind Energy Science, 2021_

## Author Comment (AC1)

**Investigation Into Boundary Layer Transition Using Wall-Resolved LES and Modeled Inflow Turbulence**

B.A. Lobo, A.P. Schaffarczyk, M. Breuer

**Review # 1**

We appreciate the effort of the reviewer for evaluating our manuscript in detail. In the following his/her remarks are answered and modifications resulting from his/her comments are explained. Note that in the annotated version of the manuscript all modifications (replacements, additions and deletions) regarding the remarks of reviewer # 1 will be highlighted in red when the upload of an annotated version is an option.

**Response to specific comments:**

- **Simulation matches the test section of the experimental study by Reichstein et al. (2019) but no comparisons have been made**
  The reviewer is right and the airfoil used for the present simulations corresponds to the test section of the experiment by Reichstein et al. (2019). However, no direct comparison of the current results with this experiment have been made since the present simulation is carried out at a Reynolds number of 100,000, whereas the experiment is conducted at a Reynolds number in the order of a few millions.

  Further information: Future simulations at higher Reynolds numbers and in comparison to that of the experiment are being carried out. The reason for running a first simulation at a relatively low Reynolds number of 100k and then stepping up the Reynolds number incrementally is because transitional studies using wall-resolved LES around airfoils for Re numbers in the order of a few millions are rarely available.

- **Comparison with XFOIL**
  The reviewer is right. A comparison with XFOIL would be helpful for serving as a benchmark to the readers. Fig. 1 (at the end of this document) shows the pressure coefficient plot with the results predicted by XFOIL included. This will be added to the next version of the manuscript.

  Note: In the current version of the manuscript there was an error in the plotting of the pressure coefficient. The reference pressure was not taken into consideration and this will be changed in the second version before publication. This does not affect any of the results or calculations other than the plot of the lift-to-drag ratio, which will also be updated. The corrected lift-to-drag plot can be seen in Fig. 2.

- **Comparison with the work of Breuer (2018) and Breuer and Schmidt (2019)**
  More information comparing the results with the previous study is a good suggestion and more information will be added to the next version of the manuscript. The following points will be added where appropriate. A short summary is as follows:

  On the relatively thinner airfoil (8.51 %) in the study by Breuer (2018), separation takes place close to the leading edge at around 20 % chord and moves downstream

with increasing turbulence intensity. Furthermore, a corresponding reduction in the chordwise extension of the separation bubble is seen before it disappears at a TI of 5.6 %. The time-averaged results showed a decrease in the drag coefficient with increasing TI. A more detailed analysis revealed that the contribution of the pressure component decreased due to the reduction in the length of the separation bubble while that of the friction component increased due to increasing inflow TI. In the current study on the flow around the thicker (20 % thickness) LM45 airfoil, the separation bubble moves slightly downstream with increasing TI before disappearing at a TI of 11.2 %. However, here the length of the separation bubble does not decrease with increasing TI. The absence of a separation bubble at a TI of 11.2 % is due to the increased momentum exchange within the boundary layer with the flow being transitional and closer to the turbulent regime than the laminar regime at the location, where it would have otherwise separated. Correspondingly, a resulting increase of the drag coefficient with increasing TI is seen.

In the study by Breuer (2018) a decrease in the lift coefficient is observed with an increase in TI up to 5.6 % before it stays constant. It is known that a separation bubble close to the leading edge could increase the lift coefficient due to the increase in the apparent camber caused by the presence of the separation bubble. With increasing TI and the downstream shift of the separation bubble, the lift coefficient then decreases. In the present study, the lift coefficient increases with increasing TI, however very slightly (a relative change of 3 %) and is likely caused by the slight downstream shift of the separation region, which increases the extent of the laminar flow along the chord.

A combination of these factors results in an increasing lift-to-drag ratio with increasing TI in Breuer (2018), whereas the lift-to-drag ratio in the current study reduces.

- **Grid resolution - Comparison of the actual values computed**
  Figure 3 of this reply shows a comparison of the pressure coefficient between the standard and the refined grid with about three times more grid points. With the refined grid the separation bubble can easily be identified by the flattened $c_p$ curve. The corresponding $c_f$ plot can be seen in Fig. 4 of this reply. Again, some deviations between the results on both grids are visible. However, for the current study which is focused on the transition phenomena, the standard grid provides a sufficiently accurate resolution with no significant changes observed in the mode of transition as seen in Fig. 1 of the manuscript. Please additionally note that the suction side is of special interest in our study and it has a finer grid resolution than the pressure side. Details are found in Table 2 of the manuscript. Taking into account the goal of the present study and the very high computational costs already necessary for the long-lasting time-consuming predictions on the standard grid, the present resolution is deemed to be sufficient for the purpose of this study.

- **High-frequency component in Fig. 1 around the mid-blade**
  The high frequency components around the mid-blade are caused by numerical noise. They are only visible near the region of breakdown to turbulence and according to our analysis do not directly affect the transition process. The cause cannot be easily isolated since in the case of an unstable system even the smallest disturbances caused by round-off errors could be a possible reason. The present simulations are carried out

with REAL*8 accuracy yielding an accuracy up to 12 or 13 digits, so this reason is unlikely. Other larger sources of disturbances are convergence errors. In the present code the Poisson equation for the pressure correction is solved based on an iterative method, i.e., the incomplete LU decomposition method of Stone (1968). The iteration is stopped if the mass conservation is fulfilled up to a convergence criterion of about $10^{-9}$. Thus, it is not exact and induces disturbances. A third possible source of disturbances is due to modeling errors by the dynamic subgrid-scale model. Finally, also some minor numerical oscillations due to the application of the central second-order accurate scheme can not be fully excluded. That scheme has the advantage of low numerical dissipation, which is important for LES. On the other hand, it is prone to numerical oscillations. Unfortunately, all these errors can not be clearly separated. Hence, it is impossible to give the real reason.

- **Addition of a grid to some figures**
  A grid or axis will be included in those images, where it is presently absent in order to improve the readability.

- **Input from Reichstein (2019) experiments for the length and time scales**
  The length and time-scales were chosen from the experiment by Hain et al. (2009) due to the similar order of magnitude of the Reynolds number. The second and more important reason for this choice is the fact that the scales from the experimental data are very large and would require a large computational domain also in the spanwise direction (up to 8 times the chosen spanwise extension) which would make the task computationally infeasible.

  Further information: For future simulations at higher Reynolds numbers as discussed above, the anisotropic inflow turbulence will be generated based on the Kaimal formulation (IEC61400-1) wherein the length/time scales are relatively determined based on a single integral scale. In this case, the scales will be determined based on the spanwise extent of the domain and not on the experimental data, again due to the computational costs being a limiting factor.

- **Clarification on what dimensionless units refer to**
  All the dimensions are non-dimensionalized with respect to the inflow velocity $u_\infty$ and the chord length $c$ as mentioned in the manuscript.

- **Figure 2 comparison to the Kolmogorov -5/3 spectrum**
  The next version of the manuscript will be updated with a plot showing the Kolmogorov -5/3 slope. This can be already seen in Fig. 5 of this reply.

- **Definition of the shape factor**
  The definition of the shape factor can be found on line 300 of the original manuscript. It is defined as the ratio between the displacement thickness and the momentum thickness.

- **Good correlation in the transition location (line 313)**
  Thanks to the reviewer for pointing this out. Line 307 needs to be explained a bit better. A corresponding change will be made to the second version of the manuscript. What is meant on line 307 is that the crucial transition onset is seen not at the point of

minimum friction coefficient $c_f$, but somewhere in-between the region where it begins to drop (around 58 % chord at a TI of 0 %) and the point of minimum $c_f$ (around 64 % chord at a TI of 0 %). This is in good correlation with the maximum of the shape factor. It must further be pointed out that these are just common methods used in the literature and it is difficult to pick a specific point as the location for the onset of transition. Therefore, the two methods are not expected to provide exactly the same chordwise position. However, we do see a very good correlation with transition being in a similar region and the change in the transition location with varying inflow turbulence also being reflected by both the $c_f$ plot and the plot of the shape factor.

- **Figure 5 uf to u′**
  Yes, that is correct. uf $= u'$. The legend will be changed to reflect this in the next version.

- **Diwan and Ramesh flat plate**
  The reviewer is right. The experimental and theoretical study by Diwan and Ramesh (2009) was conducted on a flat plate. However, it was set-up in a wind tunnel such that there was an imposed pressure gradient typical for an airfoil with separation. Due to the imposed pressure gradient the effects of curvature from a typical airfoil are seen on the flat plate. What we have learned from this experiment and our simulations is that in the region of an adverse pressure gradient (which is often caused by blade curvature) and at relatively low Reynolds numbers that allow for laminar flow separation, the region upstream of separation can already show the presence of inflectional instabilities. These instabilities are typically associated with the separated flow region due to an adverse pressure gradient but the study by Diwan and Ramesh and the present LES simulation shows that they can also be present in the attached flow, specifically, the region directly upstream of separation.

- **Figure 9f explanation**
  A short explanation of this image will be added to the manuscript around line 404 to 406. The text would change to: *"In order to follow the transition process even in the laminar separation region, data evaluations were carried out at the height of the boundary layer displacement thickness and at the mid-span. These locations have been marked and can be visualized in Fig. 9 f."*

- **Isotropic instead of atmospheric turbulence**
  Yes, the reviewer is right and the goal of the PhD project to which this simulation belongs is to ultimately study the effects of atmospheric turbulence at a high Reynolds number of about 1 million. However, as described above, the Reynolds number will be stepped-up incrementally. The second case is being run at a Reynolds number of 500k and the third case at a Re = 1M. In these cases anisotropic turbulence based on the Kaimal formulation (IEC61400-1) resembling atmospheric inflow will be used. In the present study isotropic turbulence was used to match previous studies at this Reynolds number (Hain et al. 2009, Breuer 2018, and Breuer and Schmidt 2019).

We gratefully acknowledge the effort of the referee and his/her contributions in enhancing

the quality of our paper. Thanks a lot.

B.A. Lobo, A.P. Schaffarczyk, M. Breuer

[Figure]

Figure 1: Pressure coefficient based on the averaged flow at different inflow turbulence intensities at a Reynolds number of $10^5$ and $\alpha = 4°$.

[Figure]

Figure 2: Distribution of the lift-to-drag ratio at different inflow turbulence intensities at a Re $= 10^5$ and $\alpha = 4°$. The right vertical axis shows the lift and drag coefficient scaled by the corresponding lift and drag coefficients of the reference case with a TI of 0 %.

[Figure]

Figure 3: Pressure coefficient of the standard vs. the refined grid at a TI = 0 %.

[Figure]

Figure 4: Friction coefficient of the standard vs. the refined grid at a TI = 0 %.

[Figure]

Figure 5: Turbulent kinetic energy spectra of the generated inflow turbulence.

---

## Author Response (AR1)

**Investigation Into Boundary Layer Transition Using Wall-Resolved LES and Modeled Inflow Turbulence**

B.A. Lobo, A.P. Schaffarczyk, M. Breuer

**Review # 1**

We appreciate the effort of the reviewer for evaluating our manuscript in detail. In the following his/her remarks are answered and modifications resulting from his/her comments are explained. Note that in the annotated version of the manuscript all modifications (replacements, additions and deletions) regarding the remarks of reviewer # 1 will be highlighted in red.

**Response to specific comments:**

- **Simulation matches the test section of the experimental study by Reichstein et al. (2019) but no comparisons have been made**
  The reviewer is right and the airfoil used for the present simulations corresponds to the test section of the experiment by Reichstein et al. [1]. However, no direct comparison of the current results with this experiment have been made since the present simulation is carried out at a Reynolds number of 100,000, whereas the experiment is conducted at a Reynolds number in the order of a few millions.

  Further information: Future simulations at higher Reynolds numbers and in comparison to that of the experiment are being carried out. The reason for running a first simulation at a relatively low Reynolds number of 100k and then stepping up the Reynolds number incrementally is because transitional studies using wall-resolved LES around airfoils for Re numbers in the order of a few millions are rarely available.

  Changes to manuscript: None

- **Comparison with XFOIL**
  The reviewer is right. A comparison with XFOIL would be helpful for serving as a benchmark to the readers. Fig. 1 (at the end of this document) shows the pressure coefficient plot with the results predicted by XFOIL included. This was added to the revised version of the manuscript.

  Note: In the current version of the manuscript there was an error in the plotting of the pressure coefficient. The reference pressure was not taken into consideration and this was changed in the revised version before publication. This does not affect any of the results or calculations other than the plot of the lift-to-drag ratio, which was also updated. The corrected lift-to-drag plot can be seen in Fig. 2 of this document and Fig. 5(d) of the manuscript.

  Changes to manuscript v2: Line 245 to 247 and Fig 2

  Changes to manuscript (tracked changes): Line 245 to 247 and Fig 2

- **Comparison with the work of Breuer (2018) and Breuer and Schmidt (2019)**
  More information comparing the results with the previous study is a good suggestion and more information was added to the revised version of the manuscript. The following points were added where appropriate. A short summary is as follows:

  On the relatively thinner airfoil (8.51 %) in the study by Breuer [2], separation takes place close to the leading edge at around 20 % chord and moves downstream with increasing turbulence intensity. Furthermore, a corresponding reduction in the chordwise extension of the separation bubble is seen before it disappears at a $TI$ of 5.6 %. The time-averaged results showed a decrease in the drag coefficient with increasing $TI$. A more detailed analysis revealed that the contribution of the pressure component decreased due to the reduction in the length of the separation bubble while that of the friction component increased due to increasing inflow $TI$. In the current study on the flow around the thicker (20 % thickness) LM45 airfoil, the separation bubble moves slightly downstream with increasing $TI$ before disappearing at a $TI$ of 11.2 %. However, here the length of the separation bubble does not decrease with increasing $TI$. The absence of a separation bubble at $TI = 11.2$ % is due to the increased momentum exchange within the boundary layer with the flow being transitional and closer to the turbulent regime than the laminar regime at the location, where it would have otherwise separated. Correspondingly, a resulting increase of the drag coefficient with increasing $TI$ is seen.

  In the study by Breuer [2] a decrease in the lift coefficient is observed with an increase in $TI$ up to 5.6 % before it stays constant. It is known that a separation bubble close to the leading edge could increase the lift coefficient due to the increase in the apparent camber caused by the presence of the separation bubble. With increasing $TI$ and the downstream shift of the separation bubble, the lift coefficient then decreases. In the present study, the lift coefficient increases with increasing $TI$, however, very slightly (a relative change of 3 %) and is likely caused by the slight downstream shift of the separation region, which increases the extent of the laminar flow along the chord.

  A combination of these factors results in an increasing lift-to-drag ratio with increasing $TI$ in Breuer [2], whereas the lift-to-drag ratio in the current study reduces.

  Changes to manuscript v2: Line 338 to 358

  Changes to manuscript (tracked changes): Line 339 to 359

- **Grid resolution - Comparison of the actual values computed**
  Figure 3 of this reply (Fig. 2(a) of the manuscript) shows a comparison of the pressure coefficient between the standard and the refined grid with about three times more grid points. With the refined grid the separation bubble can easily be identified by the flattened $c_p$ curve. The corresponding $c_f$ plot can be seen in Fig. 4 of this reply (Fig. 2(b) of the manuscript). Again, some deviations between the results on both grids are visible. However, for the current study, which is focused on the transition phenomena, the standard grid provides a sufficiently accurate resolution with no significant changes observed in the mode of transition as seen in Fig. 1 of the manuscript. Please additionally note that the suction side is of special interest in our study and it has a finer grid resolution than the pressure side. Details are found in Table 2 of the manuscript. Taking into account the goal of the present study and the very high

computational costs already necessary for the long-lasting time-consuming predictions on the standard grid, the present resolution is deemed to be sufficient for the purpose of this study.

Changes to manuscript v2: Line 244 to 253 and Fig. 2

Changes to manuscript (tracked changes): Line 245 to 254 and Fig. 2

- **High-frequency component in Fig. 1 around the mid-blade**
  The high frequency components around the mid-blade are caused by numerical noise. They are only visible near the region of breakdown to turbulence and according to our analysis do not directly affect the transition process. The cause can not be easily isolated since in the case of an unstable system even the smallest disturbances caused by round-off errors could be a possible reason. The present simulations are carried out with REAL*8 accuracy yielding an accuracy up to 12 or 13 digits, so this reason is unlikely. Other larger sources of disturbances are convergence errors. In the present code the Poisson equation for the pressure correction is solved based on an iterative method, i.e., the incomplete LU decomposition method of Stone [3]. The iteration is stopped if the mass conservation is fulfilled up to a convergence criterion of about $10^{-9}$. Thus, it is not exact and induces disturbances. A third possible source of disturbances is due to modeling errors by the dynamic subgrid-scale model. Finally, also some minor numerical oscillations due to the application of the central second-order accurate scheme can not be fully excluded. That scheme has the advantage of low numerical dissipation, which is important for LES. On the other hand, it is prone to numerical oscillations. Unfortunately, all these errors can not be clearly separated. Hence, it is impossible to give the real reason.

  Changes to manuscript: None

- **Addition of a grid to some figures**
  A grid or axis was included in those images, where it is presently absent in order to improve the readability.

  Changes to manuscript: Missing grid on Fig. 1 has been included

- **Input from Reichstein (2019) experiments for the length and time scales**
  The length and time scales were chosen from the experiment by Hain et al. [4] due to the similar order of magnitude of the Reynolds number. The second and more important reason for this choice is the fact that the scales from the experimental data are very large and would require a large computational domain also in the spanwise direction (up to 8 times the chosen spanwise extension) which would make the task computationally infeasible.

  Further information: For future simulations at higher Reynolds numbers as discussed above, anisotropic inflow turbulence will be generated based on the Kaimal formulation (IEC61400-1) wherein the length/time scales are relatively determined based on a single integral scale. In this case, the scales will be determined based on the spanwise extent of the domain and not on the experimental data, again due to the computational costs being a limiting factor.

  Changes to manuscript: None

- **Clarification on what dimensionless units refer to**
  All the dimensions are non-dimensionalized with respect to the inflow velocity $u_\infty$ and the chord length $c$ as mentioned in the manuscript.

  Changes to manuscript: None

- **Figure 2 comparison to the Kolmogorov -5/3 spectrum**
  The revised version of the manuscript was updated with a plot showing the Kolmogorov -5/3 slope. This can be already seen in Fig. 5 of this reply.

  Changes to manuscript: Fig. 3(a) has been updated

- **Definition of the shape factor**
  The definition of the shape factor can be found on line 300 of the original manuscript (line 310 on the updated manuscript). It is defined as the ratio between the displacement thickness and the momentum thickness.

- **Good correlation in the transition location (line 313)**
  Thanks to the reviewer for pointing this out. Line 307 (of the original manuscript) needs to be explained a bit better. A corresponding change was made to the revised version of the manuscript. What is meant on line 307 is that the crucial transition onset is seen not at the point of minimum friction coefficient $c_f$, but somewhere in-between the region where it begins to drop (around 58 % chord at $TI = 0$ %) and the point of minimum $c_f$ (around 64 % chord at $TI = 0$ %). This is in good correlation with the maximum of the shape factor. It must further be pointed out that these are just common methods used in the literature and it is difficult to pick a specific point as the location for the onset of transition. Therefore, the two methods are not expected to provide exactly the same chordwise position. However, we do see a very good correlation with transition being in a similar region and the change in the transition location with varying inflow turbulence also being reflected by both the $c_f$ plot and the plot of the shape factor.

  Changes to manuscript v2: Line 324 to 325 and slight changes to sentence structure on line 328 to 330

  Changes to manuscript (tracked changes): Line 325 to 326 and slight changes to sentence structure on line 329 to 331

- **Figure 5 uf to u'**
  Yes, that is correct. uf = $u'$. The legend was changed to reflect this in the revised version.

  Changes to manuscript: Fig. 6 has been updated

- **Diwan and Ramesh flat plate**
  The reviewer is right. The experimental and theoretical study by Diwan and Ramesh [5] was conducted on a flat plate. However, it was set-up in a wind tunnel such that there was an imposed pressure gradient typical for an airfoil with separation. Due to the imposed pressure gradient, the effects of curvature from a typical airfoil are seen on the flat plate. What we have learned from this experiment and our simulations is that in the region of an adverse pressure gradient (which is often caused by blade curvature)

and at relatively low Reynolds numbers that allow for laminar flow separation, the region upstream of separation can already show the presence of inflectional instabilities. These instabilities are typically associated with the separated flow region due to an adverse pressure gradient but the study by Diwan and Ramesh [5] and the present LES simulation shows that they can also be present in the attached flow, specifically, the region directly upstream of separation.

Changes to manuscript: None

- **Figure 9 f explanation**
A short explanation of this image was added to the manuscript. The text was change to: *"In order to follow the transition process even in the laminar separation region, data evaluations were carried out at the height of the boundary layer displacement thickness and at the mid-span. These locations have been marked and can be visualized in Fig. 10 f."*

Changes to manuscript v2: Line 443

Changes to manuscript (tracked changes): Line 444

- **Isotropic instead of atmospheric turbulence**
Yes, the reviewer is right and the goal of the PhD project to which this simulation belongs, is to ultimately study the effects of atmospheric turbulence at a high Reynolds number of about 1 million. However, as described above, the Reynolds number will be stepped up incrementally. The second case is being run at a Reynolds number of 500k and the third case at Re = 1M. In these cases anisotropic turbulence based on the Kaimal formulation (IEC61400-1) resembling atmospheric inflow will be used. In the present study isotropic turbulence was used to match previous studies at this Reynolds number (Hain et al. [4], Breuer [2], and Breuer and Schmidt [6]).

Changes to manuscript: None

**Review # 2**

We appreciate the effort of the reviewer for evaluating our manuscript in detail. In the following his/her remarks are answered and modifications resulting from his/her comments are explained. Note that in the annotated version of the manuscript all modifications (replacements, additions and deletions) regarding the remarks of reviewer # 2 will be highlighted in blue. In case the remarks coincide with that of reviewer 1, the changes may be found in red

**Response to specific comments:**

- **More detailed grid refinement study**
  Figure 3 of this reply (Fig. 2(a) of the manuscript) shows a comparison of the pressure coefficient between the standard and the refined grid with about three times more grid points. With the refined grid the separation bubble can easily be identified by the flattened $c_p$ curve. The corresponding $c_f$ plot can be seen in Fig. 4 of this reply (Fig. 2(b) of the manuscript). Again, some deviations between the results on both grids are visible. However, for the current study, which is focused on the transition phenomena, the standard grid provides a sufficiently accurate resolution with no significant changes observed in the mode of transition as seen in Fig. 1 of the manuscript. Please additionally note that the suction side is of special interest in our study and it has a finer grid resolution than the pressure side. Details are found in Table 2 of the manuscript. Taking into account the goal of the present study and the very high computational costs already necessary for the long-lasting time-consuming predictions on the standard grid, the present resolution is deemed to be sufficient for the purpose of this study.

  In principle it would be possible to include a study of the resolved vs. the modeled turbulence. However, the manuscript is already quite long and the addition of this feature would not provide a lot of useful information to the reader and would also take some time for the computations to be performed. Nevertheless, this is a valuable suggestion and would be something to consider for the higher Re cases. Note that in a recent study by Solís-Gallego et al [7] who have also performed a wall-resolved LES with a grid whose dimensions are coarser than our own standard grid and who have also used the dynamic Smagorinsky subgrid-scale model has investigated this topic. They found that a very small percentage of the turbulent kinetic energy is modeled (between 1 and 8 %) which is below their acceptable threshold of 20 %.

  Changes to manuscript v2: Line 244 to 253 and Fig. 2

  Changes to manuscript (tracked changes): Line 245 to 254 and Fig. 2

- **Geometry, mesh set-up and boundary conditions**
  Yes, the reviewer is right and the angle of attack is already included in the basic mesh set-up. Therefore, there is no need of any special outflow conditions on the upper boundary.

  The domain extends 8 chord lengths upstream of the airfoil and 15 chord lengths downstream of the airfoil. With increasing distance from the airfoil, the grid grows coarser (geometric expansion of the grid spacing). This damps out interfering waves. In addition to the damping due to the grid coarsening, the convective boundary condition

on the outflow plane (see line 262 of the manuscript) has proven itself for LES as it avoids the problem of reflection of pressure waves at the outflow edge and any associated error propagation back into the inner integration domain (see, e.g., [8, 9, 10]). Furthermore, note that there have been various studies that use similar domain dimensions (see, e.g., [7, 11, 12, 13]).

Changes to manuscript: Line 209

- **Decay of turbulence intensity**
  Definition of the turbulence intensity:
  The decay of free-stream turbulence is plotted using the averaged Reynolds stresses at the end of the simulation period defined as $TI = \sqrt{\frac{1}{3} \times (u'u' + v'v' + w'w')}$, where $u'u'$, etc. are the averaged normal Reynolds stresses in the three principle directions. Future studies will include monitoring points so that the turbulence statistics could be more accurately calculated instead of depending on the averaged statistics.

  Where is the required $TI$ expected to be achieved:
  The required turbulence intensity is expected to be achieved slightly upstream of the airfoil. An analysis of the development of the inflow turbulence was conducted by Breuer [2] for the same inflow turbulence conditions. A development length of about one chord length was found to be sufficient depending on the turbulent length scale. In the present case, the fully developed turbulence is seen at around $x/c = $ -1 to -0.2, which is just upstream of the region influenced by the airfoil.

  Simulations in an empty field for calibration:
  No, simulations without the wing in an empty domain have not been conducted for the purpose of calibration. However, as seen from the plot generated from the averaged Reynolds stresses, it is clear that the $TI$ achieved is of the similar order as that expected, albeit slightly lower which could be due to the averaging of the Reynolds stresses which also includes data prior to the inclusion of inflow turbulence. It must be pointed out that the absolute value of the turbulence intensity is not vital as the simulations study the transition scenario with increasing turbulence. Future studies which include monitoring points will be more accurate.

  Peak in turbulence at the start:
  The peak seen around $x/c = $ -2 (the location of the inflow plane) is due to the way in which the turbulence is injected into the domain. To reduce the required development length of the synthetic turbulence and to avoid discontinuities, the source terms are superimposed in a predefined influence area equal to twice the length scale of the injected turbulence. The center of this influence area is the inflow plane. Based on a Gaussian bell-shaped distribution the source terms are then scaled within this influence area. Further details on the procedure are found in Breuer [2].

  Variations in the data:
  Yes, the data are statistically converged for the mean flow around the airfoil ($c_p$ and $c_f$ was monitored). In all cases, the airfoil seems to influence the expected $TI$ beginning at around $x/c = $ -0.2. An increase in effective $TI$ up to the separation/transition point (around 50 % chord) is observed before it begins to drop. In the case of $TI = 11.2$ % there is no separation bubble and this seems to be the reason why there is a continuous

decay in turbulence (after the small increase above the leading edge of the airfoil as also seen in the other cases). It probably has something to do with the boundary layer being thinner in this case due to the absence of a separation bubble and the fact that the analysis for the calculation of $TI$ is conducted at a height corresponding to the boundary layer thickness at 50 % chord for the $TI = 0$ % case.

Changes to manuscript: None

- **Comparisons between other studies featuring the SD7003 and NACA0018 airfoils**
  More information comparing the results with other studies is a good suggestion and more information was added to the revised version of the manuscript where appropriate. A short summary is as follows:

On the relatively thinner airfoil (SD7003) (8.51 %) in the study by Breuer [2], separation takes place close to the leading edge at around 20 % chord and moves downstream with increasing turbulence intensity. Furthermore, a corresponding reduction in the chordwise extension of the separation bubble is seen before it disappears at $TI = 5.6$ %. The time-averaged results showed a decrease in the drag coefficient with increasing $TI$. A more detailed analysis revealed that the contribution of the pressure component decreased due to the reduction in the length of the separation bubble while that of the friction component increased due to increasing inflow $TI$. In the current study on the flow around the thicker (20 % thickness) LM45 airfoil, the separation bubble moves slightly downstream with increasing $TI$ before disappearing at $TI = 11.2$ %. However, here the length of the separation bubble does not decrease with increasing $TI$. The absence of a separation bubble at $TI = 11.2$ % is due to the increased momentum exchange within the boundary layer with the flow being transitional and closer to the turbulent regime than the laminar regime at the location, where it would have otherwise separated. Correspondingly, a resulting increase of the drag coefficient with increasing $TI$ is seen.

In the study by Breuer [2] a decrease in the lift coefficient is observed with an increase in $TI$ up to 5.6 % before it stays constant. It is known that a separation bubble close to the leading edge could increase the lift coefficient due to the increase in the apparent camber caused by the presence of the separation bubble. With increasing TI and the downstream shift of the separation bubble, the lift coefficient then decreases. In the present study, the lift coefficient increases with increasing $TI$, however very slightly (a relative change of 3 %) and is likely caused by the slight downstream shift of the separation region, which increases the extent of the laminar flow along the chord.

A combination of these factors results in an increasing lift-to-drag ratio with increasing $TI$ in Breuer [2], whereas the lift-to-drag ratio in the current study reduces.

Tangermann and Klein [14] have studied the influence of inflow turbulence with varying intensities and length scales on the NACA0018 airfoil. In this case the separation bubble already originates between 25 and 35% chord length which is further downstream than in our simulations. This can be attributed to the location of the maximum thickness of the airfoil since the adverse pressure gradient favors separation. In case of the NACA0018 airfoil, its maximum thickness is located at 30 % chord while the airfoil studied here has its maximum thickness located at 36 % chord. Tangermann

and Klein [14] have also observed the influence of inflow turbulence on the separation bubble with the separation region being delayed in some spanwise regions compared to the case without added turbulence.

Changes to manuscript v2: Line 317 to 322, 338 to 358 and 524 to 525.

Changes to manuscript (tracked changes): Line 316 to 321, 339 to 359 and 523 to 524.

- **Brief discussion on the instantaneous state at a $TI$ of 11.2 %**
  There seems to be a misunderstanding. It probably arises on account of line 66 of the first version of the manuscript which states *"The percentage of time, where the spanwise averaged flow on the suction surface is attached, was found to be 59.8 % at a $TI$ of 8 % and 96.6 % at a $TI$ of 10 % which indicates that there were instances of separation even at a high $TI$ of 10 %"*. This statement corresponds to the investigation by Zaki et al. [15].

  In our study we do not see separation either in the mean or the instantaneous flow field, but the data was not as critically analyzed as in the investigation by Zaki et al. [15] to look for possible and very short occurrences of instantaneous flow separation. On line 305 of the first version of the manuscript, it is stated that *"at $TI = 11.2$ % the separation bubble vanishes"* and on line 352 *"For the case with a very high turbulence intensity of 11.2 %, the flow does not separate and spanwise rolls are no longer present while the streaks take over the transition process."* and finally on line 527 this is again stated in the conclusion.

  The manuscript already discusses the instantaneous transition phenomenon at $TI = 11.2$ % with transition being dominated by Klebanoff modes (boundary layer streaks). No further discussion is necessary since the interaction of streaks and the varicose mode of transition has already been thoroughly discussed in Section 4.4 using the case with an inflow $TI$ of 2.8 % as reference. Similar processes are observed at higher inflow turbulence intensities, albeit with relatively more streaks being formed within the boundary layer.

  Changes to manuscript: None

- **Specific comments**
  Figure 1 has been updated and the other errors pointed out have been corrected.

  Changes to manuscript: Fig. 1 has been updated and the separation zone is indicated.

- **Correction to $c_p$**
  In the first version of the manuscript there was an error in the plotting of the pressure coefficient. The reference pressure was not taken into account and this has been changed in the second version. This does not affect any of the results or calculations other than the plot of the lift-to-drag ratio, which has also been updated.

We gratefully acknowledge the effort of the referees and his/her contributions in enhancing the quality of our paper. Thanks a lot.

B.A. Lobo, A.P. Schaffarczyk, M. Breuer

**References**

[1] Torben Reichstein, Alois Peter Schaffarczyk, Christoph Dollinger, Nicolas Balaresque, Erich Schülein, Clemens Jauch, and Andreas Fischer. Investigation of laminar-turbulent transition on a rotating wind-turbine blade of multi-megawatt class with thermography and microphone array. *Energies*, 12(11):2102, 2019.

[2] M. Breuer. Effect of inflow turbulence on an airfoil flow with laminar separation bubble: An LES study. *J. Flow, Turbulence and Combustion*, 101(2):433–456, 2018.

[3] H. L. Stone. Iterative solution of implicit approximations of multidimensional partial differential equations. *SIAM J. Num. Anal.*, 5:530–558, 1968.

[4] R. Hain, C. J. Kähler, and R. Radespiel. Dynamics of laminar separation bubbles at low–Reynolds number aerofoils. *J. Fluid Mech.*, 630:129–153, 2009.

[5] S. S. Diwan and O. N. Ramesh. On the origin of the inflectional instability of a laminar separation bubble. *J. Fluid Mech.*, 629:263–298, 2009.

[6] M. Breuer and S. Schmidt. Effect of inflow turbulence on LES of an airfoil flow with laminar separation bubble. In M. V. Salvetti, V. Armenio, J. Fröhlich, B. J. Geurts, and H. Kuerten, editors, *ERCOFTAC Series, Direct and Large-Eddy Simulation XI, 11th Int. ERCOFTAC Workshop on Direct and Large–Eddy Simulation: DLES-11, Pisa, Italy, May 29–31, 2017*, volume 25, pages 351–357. Springer Nature Switzerland AG, 2019.

[7] I. Solís-Gallego, K. M. Argüelles Díaz, J. M. Fernández Oro, and S. Velarde-Suárez. Wall-resolved LES modeling of a wind turbine airfoil at different angles of attack. *J. Mar. Sci. Eng.*, 8(3):212, 2020.

[8] M. Breuer and M. Pourquié. First experiences with LES of flows past bluff bodies. In W. Rodi and G. Bergeles, editors, *Engineering Turbulence Modelling and Experiments 3, 3rd Int. Symp. on Engineering Turbulence Modelling and Measurements, May 27–29, 1996*, pages 177–186, Heraklion, Crete, Greece, 1996. Elsevier.

[9] M. Breuer. Numerical and modeling influences on large–eddy simulations for the flow past a circular cylinder. *Int. J. Heat Fluid Flow*, 19(5):512–521, 1998.

[10] M. Breuer. A challenging test case for large–eddy simulation: High Reynolds number circular cylinder flow. *Int. J. Heat Fluid Flow*, 21(5):648–654, 2000.

[11] Ch. P. Mellen, J. Fröhlich, and W. Rodi. Lessons from LESFOIL project on large-eddy simulation of flow around an airfoil. *AIAA J*, 41(4):573–581, 2003.

[12] J. Ke and J. R. Edwards. Numerical simulations of turbulent flow over airfoils near and during static stall. *J. Aircr.*, 54(5):1960–1978, 2017.

[13] W. Gao, W. Zhang, W. Cheng, and R. Samtaney. Wall-modelled large-eddy simulation of turbulent flow past airfoils. *J. Fluid Mech.*, 873:174–210, 2019.

[14] E. Tangermann and M. Klein. Numerical simulation of laminar separation on a NACA0018 airfoil in freestream turbulence. In *AIAA Scitech 2020 Forum*. American Institute of Aeronautics and Astronautics, 2020.

[15] T. A. Zaki, J. G. Wissink, W. Rodi, and P. A. Durbin. Direct numerical simulations of transition in a compressor cascade: The influence of free-stream turbulence. *J. Fluid Mech.*, 665:57–98, 2010.

[Figure]

Figure 1: Pressure coefficient based on the averaged flow at different inflow turbulence intensities at a Reynolds number of $10^5$ and $\alpha = 4°$.

[Figure]

Figure 2: Distribution of the lift-to-drag ratio at different inflow turbulence intensities at a $Re = 10^5$ and $\alpha = 4°$. The right vertical axis shows the lift and drag coefficient scaled by the corresponding lift and drag coefficients of the reference case with $TI = 0$ %.

[Figure]

Figure 3: Pressure coefficient of the standard vs. the refined grid at a $TI = 0$ %.

[Figure]

Figure 4: Friction coefficient of the standard vs. the refined grid at a $TI = 0$ %.

[Figure]

Figure 5: Turbulent kinetic energy spectra of the generated inflow turbulence.

---

## Author Response (AR2)

Response to the Review on the Paper wes-2021-30

**Investigation Into Boundary Layer Transition Using Wall-Resolved LES and Modeled Inflow Turbulence**

B.A. Lobo, A.P. Schaffarczyk, M. Breuer

**Review**

We appreciate the effort of the reviewer for evaluating our manuscript in detail. In the following his/her remarks are answered and modifications resulting from his/her comments are explained. Note that in the annotated version of the manuscript all modifications (replacements, additions and deletions) regarding the remarks of the reviewer will be highlighted in red when the upload of an annotated version is an option.

**Response to specific comments:**

- **Include information from the responses in the updated manuscript**
  The reviewer is right! In order to make the manuscript self-containing, all relevant remarks from the previous reviewer responses have now been added to the manuscript.

- **Abstract as a stand-alone text**
  The statement "first step of this objective" has been removed from the abstract which now makes it stand-alone. This information continues to be available to interested readers in Section 3.1 (Description of the flow case).

- **Grid resolution**
  **Streamwise oriented structures:**
  The high frequency streamwise components seen around the separation region are caused by numerical noise. They are only visible near the region of breakdown to turbulence and according to our analysis do not directly affect the transition process which is the focus of our study. The cause was found to be some minor numerical oscillations due to the application of the central second-order accurate scheme. This scheme has the advantage of low numerical dissipation, which is important for LES. On the other hand, it is prone to numerical oscillations. Furthermore, this issue, albeit minor, is strongest in the case without added inflow turbulence. Therefore, the case of $TI = 0$ % was additionally simulated using a blended scheme with 98 % central differencing and an upwind scheme (2 %). The extra plots have been added to the manuscript and a better agreement between the standard and refined grid is seen in the $c_p$ and $c_f$ plots.

  **Grid resolution affecting $c_p$:**
  As detailed above, using a blended 2 % upwind scheme a closer match between the $c_p$ and $c_f$ plots of the predicted data on both grids is found, especially on the suction side and beyond the onset of the adverse pressure gradient region. The deviation in the laminar region of the airfoil, as has now been discussed in the manuscript, is very likely due to a geometrical issue of the airfoil smoothness. The peak in the $c_f$ plot at around 10 % chord is similar to what is seen in preliminary studies for a Reynolds number of

500k of the same airfoil. This deviation is caused by the airfoil geometry not being sufficiently smooth, an issue that becomes increasingly prominent with increasing grid resolution. By fixing the airfoil smoothing issue, the case at Re = 500k experiences an increase in the favorable pressure gradient and a smoothening of the $c_f$ distribution. It is very likely that the same issue is at play on the refined grid at Re = 100k.

**Fig. 5 and Table 3:**
The shape factor obtained from the refined grid has now been added to Fig. 5 and details on separation and reattachment have been added to Table 3. From the newly added plot of the displacement and momentum thicknesses (Figs. 5e and 5f of the updated manuscript) it is obvious that the boundary layer properties in the laminar region converge quite well, further indicating that the $c_p$ and $c_f$ distributions found on the blade surface arise due to airfoil smoothing issues. The shape factor from the standard grid with 98 % CDS and the refined grid agree quite well, but a clear discrepancy between the predicted data on the standard grid at 100 % CDS and 98 % CDS is visible. This is a result of the amplification of small variations in the displacement and momentum thicknesses on account of the way in which the shape factor is calculated. However, the location of the separation bubble (see Table 3 of the manuscript) and the corresponding location of transition onset indicated by the peak in the shape factor match quite well for these cases. As discussed in [1], the grid resolution of the standard grid is sufficient for the study of transition including separation bubbles, but a finer grid resolution could better capture the vortex development. This explains the slight difference in the shape factor between the standard and the refined grid within the region of the separation bubble. However, this does not affect the mode of transition.

**Comparison of grid resolution:**
Table 2 of the manuscript shows the grid parameters used for both the standard and the refined grid. Section 3.2.1 also compares the grid resolution to the recommendations proposed by Piomelli et al. [2]. The updated version of the manuscript also includes the suggested grid resolution from the study by Asada et al. [1] for flows involving separation bubbles. The standard grid employed for the simulations is well within the suggested parameter ranges for studies of transitional flows.

- **Additional Comments**
  To further support our simulations, in addition to the references already included, an experimental study by Boutilier et al. [3] is now added to the manuscript in which the frequency range of the separated shear layer also at a Reynolds number of 100k and a similar angle of attack on the NACA 0018 matches that of our simulations at 50 % chord in the absence of added inflow turbulence.

  **Line 325:**
  The crucial transition onset between 58 and 64 % chord refers to the standard grid and not the refined grid. It is now made clear that the results section only refers to the results from the standard grid. The refined grid is used solely for the purpose of the grid refinement study.

  **Other minor points:**
  Thank you for the comments regarding Figures 10 and 11. They have been taken into account.

We gratefully acknowledge the effort of the referee and his/her contributions in enhancing the quality of our paper. Thanks a lot.

B.A. Lobo, A.P. Schaffarczyk, M. Breuer

**References**

[1] K. Asada and S. Kawai. Large-eddy simulation of airfoil flow near stall condition at Reynolds number 2.1 $\times 10^6$. *Phys. Fluids*, 30(8):1139–1145, 2018.

[2] U. Piomelli and J. R. Chasnov. Large–eddy simulations: Theory and applications. In M. Hallbäck, D.S. Henningson, A.V. Johansson, and P.H. Alfredson, editors, *Turbulence and Transition Modeling*, pages 269–331. Kluwer, 1996.

[3] M. S. H. Boutilier and S. Yarusevych. Separated shear layer transition over an airfoil at a low Reynolds number. *Phys. Fluids*, 24(8):084105, 2012.

---

## Author Response (AR3)

Response to the Review on the Paper wes-2021-30

**Investigation Into Boundary Layer Transition Using Wall-Resolved LES and Modeled Inflow Turbulence**

B.A. Lobo, A.P. Schaffarczyk, M. Breuer

**Review**

We appreciate the effort of the reviewer for evaluating our manuscript in detail. In the following his/her remarks are answered and modifications resulting from his/her comments are explained. Note that in the annotated version of the manuscript all modifications (replacements, additions and deletions) regarding the remarks of the reviewer will be highlighted in red.

**Response to specific comments:**

- **line 200: This deviation is caused by the airfoil geometry not being sufficiently smooth**
  The issue here is not clear, but we believe it may be regarding the clarity of the statement. Therefore, the deviation we refer to has been now clearly described.

- **Figure 4: Please clarify in the caption for which case the figure is (check for other figures whether this would be required).**
  The caption of this figure already includes the relevant case to which it belongs (TI = 1.4 %) and so there was no need to add any further information. All images were checked to ensure the case it belonged to was clear.

- **Section 4.1; Figure 5: The 98 % CDS results have been added to the figure, but are not yet discussed. Can you add this discussion?**

  **Around line 440: same as the point above. Please update the discussion to include the new 98 % CDS results.**
  The above two points refer to the CDS 98 % case which we included mainly for the grid convergence study. A discussion of the results from the CDS 98 % case are therefore already included in Section 3.2.1 and there is no need to discuss them once again in Section 4.1. A discussion in Section 4.1 would only lead to confusion for the reader as we have stated that all the discussions in the results section refer to the CDS 100 % scheme. Since the plots in Fig. 5 do have the CDS 98 % data as well, a further line in the start of the results section that stresses the fact that a discussion of these has already been done in Section 3.2.1 has now been included. This should prevent any confusion to the reader.

We gratefully acknowledge the effort of the referee and his/her contributions in enhancing the quality of our paper. Thanks a lot.

B.A. Lobo, A.P. Schaffarczyk, M. Breuer